# Evaluation of Aeolus L2B wind product with wind profiling radar measurements and numerical weather prediction model equivalents over Australia

Haichen Zuo[1], Charlotte Bay Hasager[1], Ioanna Karagali[2], Ad Stoffelen[3], Gert-Jan Marseille[3], Jos de Kloe[3]

[1]Wind Energy, Technical University of Denmark, Roskilde, 4000, Denmark
[2] Danish Meteorological Institute, Copenhagen, 2100, Denmark
[3] Royal Netherlands Meteorological Institute, De Bilt, 3731 GA, Netherlands

*Correspondence to*: Haichen Zuo (hazu@dtu.dk)

**Abstract.** Carrying a laser Doppler instrument, the Aeolus satellite was launched in 2018, becoming the first mission for atmospheric wind profile measurements from space. Before utilizing the Aeolus winds for different applications, evaluating their data quality is essential. With the help of ground-based wind profiling radar measurements and the European Centre for Medium-Range Weather Forecasts (ECMWF) model equivalents, this study quantifies the error characteristics of Aeolus L2B (baseline-11) near real time horizontal line-of-sight winds across Australia during October 2020 – March 2021 by using both inter-comparison and triple collocation analysis. The results of the inter-comparison analysis indicate that both Rayleigh-clear winds and Mie-cloudy winds are in good agreement with the ground-based radar measurements with overall absolute mean biases smaller than 0.7 m s$^{-1}$ and correlation coefficients larger than or equal to 0.9. Moreover, assuming the radar measurements as the reference data set, Mie-cloudy winds are shown to be more precise than Rayleigh-clear winds with an overall random error of 4.14 m s$^{-1}$ and 5.81 m s$^{-1}$, respectively. Similar results were also found from triple collocation analysis, with error standard deviations of 5.61 m s$^{-1}$ and 3.50 m s$^{-1}$ for Rayleigh-clear winds and Mie-cloudy winds. In addition, the Mie channel is shown to be more capable of capturing the wind in the planetary boundary layer (<1,500 m). The findings of this study demonstrate the good performance of space-borne Doppler lidar for wind profiling and provide valuable information for data assimilation in numerical weather prediction.

## 1 Introduction

The lack of wind profiles is still one of the major deficiencies in the Global Observing System (GOS), which limits our knowledge of atmospheric dynamics and the performance of numerical weather prediction (NWP) (World Meteorological Organization (WMO), 2004). To help close this gap, after more than 15 years of design, the Aeolus satellite carrying an Atmospheric Laser Doppler Instrument (ALADIN) was launched by the European Space Agency (ESA) in 2018, becoming the first satellite mission in the world for measuring wind profiles from space. After a successful launch, Aeolus is in a sun-synchronous orbit with a 7-day repeat cycle. It crosses the equator at 18:00 LT (Local time) during ascending orbits (from

south to north) and at 06:00 LT during descending orbits (from north to south). The azimuth angle of Aeolus is ~260° for ascending orbits and ~100° for descending orbits, away from the poles. The viewing angle of ALADIN toward the atmosphere is 35° off-nadir. The measured wind along the laser beam line-of-sight (LOS) is then converted to the horizontal to give the horizontal line-of-sight (HLOS) wind component, which is approximately east-west oriented for most of the orbits (Andersson et al., 2008).

Wind retrievals of ALADIN are based on light scattering by atmospheric molecules and particulates (aerosol, cloud droplets, and ice crystals), which move with the ambient wind, and on the Doppler effect (Andersson et al., 2008). The laser system of ALADIN emits a beam of powerful light in the ultraviolet part of the electromagnetic spectrum at 355 nm towards the Earth. Then, the backscattered light from the atmosphere is collected by the telescope and transferred to the receiver for analysis. Since the laser light can be backscattered by both molecules and particulates in the atmosphere, ALADIN has two separate detection channels. One is for Rayleigh scattering from molecules, such as oxygen and nitrogen, with the diameter being about 0.3-0.4 nm that is smaller than the light wavelength; the other is for Mie scattering from the large particles such as cloud droplets and ice crystals, dust, and aerosols, the diameters of which are usually greater than 1 $\mu m$ (Calvert, 1990; Wallace and Hobbs, 2006b; Ingmann and Straume, 2016; Vallejos-Burgos et al., 2018). From the backscattered signal, winds from the surface to about 30 km in height can be derived, depending on the range bin settings, i.e., the size of the 24 bins defining the wind profile.

By detecting global wind profiles from space, the Aeolus satellite has the potential to serve a variety of applications, including further exploring atmospheric dynamics, improving numerical weather predictions, and better estimating the dispersion of air pollutants (Banyard et al., 2021; ESA, 2021; Rennie et al., 2021). However, before employing Aeolus winds for different applications, it is essential to know the error characteristics of the wind products. In-situ measurements (e.g. radiosondes), ground-based remote sensing observations (e.g. lidar or radar) and NWP model equivalents are the three main reference products used for wind validation. After the successful launch, Aeolus winds have been inter-compared with different reference data over many regions. For example, Aeolus winds in the early mission stage were compared with radiosonde observations in different climate zones over the Atlantic Ocean and NWP model equivalents for the Northern Hemisphere, and larger biases were reported for both Rayleigh-clear and Mie-cloudy winds (Baars et al., 2020; Martin et al., 2021). This is associated with the early processing algorithms which have since been developed further to account for such issues. Later, Guo et al. (2021) compared the Aeolus winds with radar wind profiler (RWP) measurements over China, showing that the mean differences are -0.64 m s$^{-1}$ and -0.28 m s$^{-1}$ with the standard deviations of 6.82 m s$^{-1}$ and 4.2 m s$^{-1}$ for Rayleigh-clear and Mie-cloudy winds, respectively. Validation was also conducted over the polar regions, and a good agreement with ground-based RWP measurements was obtained in most cases. However, a significant bias of around 7 m s$^{-1}$ during the ascending orbits and a larger random error were found for the Mie channel detection in Antarctica, associated with the sunlight scattering in summer (Belova et al., 2021). More recently, Iwai et al. (2021) validated Aeolus 2B02 and 2B10 wind products by comparing with

wind profilers, ground-based coherent Doppler wind lidars and GPS radiosondes over Japan, with the inter-comparison results for wind profilers and radiosondes showing improved quality of Aeolus 2B10 winds as both biases and random errors were smaller compared to those for 2B02 product. Although validation and calibration have been carried out over many regions, the data quality of Aeolus measurements across Australia has not been investigated so far.

Moreover, regarding the validation method, most works related to Aeolus are based on inter-comparison analysis. In addition to this, triple collocation analysis is another advantageous method to evaluate space-borne remote sensing products. Unlike inter-comparison that treats a reference data set free of errors, triple collocation analysis requires three independent measurement systems and assumes that each system contributes to the truth. The outputs are the error standard deviation of each system and calibration relations based on a reference data set, which can provide valuable information for cost function in data assimilation (Stoffelen, 1998; Vogelzang et al., 2011). Triple collocation analysis has been widely employed to assess the wind measurements from different instruments, including scatterometers, altimeters and radiometers (Caires, 2003; Portabella and Stoffelen, 2009; Ribal and Young, 2020). However, very few studies have evaluated wind products from the space-borne lidar by this method so far. To complement to earlier validation studies, this study evaluates the Aeolus L2B HLOS wind product over Australia by inter-comparison with ground-based wind profiling radar (WPR) measurements. In addition, a triple collocation analysis for Aeolus HLOS winds is conducted with the help of WPR measurements and NWP model equivalents.

A description of the data and methods used in this study is available in Sect. 2. Key research findings from data analysis are presented in Sect. 3, followed by the discussions in Sect. 4. The final Sect. summarizes the study briefly and draws conclusions.

## 2 Data and methods

### 2.1 Aeolus L2B wind product

Aeolus Level-2B baseline 11 near real time HLOS winds during the Australian summer from October 2020 to March 2021 were used for validation, being the most recent available near real time wind product when conducting this study. The data were obtained from the ESA Aeolus Online Dissemination System (http://aeolus-ds.eo.esa.int/oads/access/) (ESA, 2021). According to signal-to-noise ratio, Aeolus L2B winds are categorised into four types, which are Rayleigh-clear, Rayleigh-cloudy, Mie-clear and Mie-cloudy (de Kloe et al., 2021). The measurements from the Rayleigh channel have better performance in a clear sky (Rayleigh-clear), for which there is little or no contamination from Mie scattering; the wind measurements in the Mie channel need strong backscattering from aerosols, water droplets or ice crystals (Mie-cloudy) (Rennie et al., 2020). In addition, Rayleigh-clear and Mie-cloudy winds are currently the only two types of Aeolus winds that are assimilated into the ECMWF model for operational weather forecast (Rennie et al., 2021). Based on these considerations, only Rayleigh-clear and Mie-cloudy winds were extracted for evaluation. The horizontal accumulation along the ground track of

each observation is typically 87 km for Rayleigh winds (which takes 12 seconds) and 15 km for Mie winds (which takes around 2 seconds), but it may be shorter locally due to the classification in cloudy and clear scenes. Vertically, there are 24 range bins with sizes varying from 250 m to 2 km. To capture the characteristics of atmospheric circulation over different climate zones, range bin settings vary along the orbit according to geographic location and in time, as requested by special measurement requests and to adapt to seasons and climate zones. Over Australia, there are two different range bin setting regions (Fig.1), i.e., the tropical setting region (30°S - 30°N) and the extratropical setting region (30°S - 60°S). The differences in range bin settings are measurement heights and range bin thickness. For tropical setting, the measurements can reach just over 20 km in height with a range bin thickness of 750 m between 12 km and 15 km to capture the gravity waves, while the maximum measurement height of the extratropical setting is about 17.5 km with a higher vertical resolution of 500 m between 5 km and 10 km for jet stream detection (ESA, 2020).

Referring to the existing recommendations for quality control, the HLOS wind speed with a validity flag of 0 and estimated error larger than 8 m s$^{-1}$ for Rayleigh-clear winds and 4 m s$^{-1}$ for Mie-cloudy winds were removed (Rennie and Isaksen, 2020).

## 2.2 Wind profiling radar measurements

Wind profiling radar (WPR) is one of the remote sensing equipment that can measure the three-dimensional wind field (Dolman et al., 2018). The Australian WPR network is operated by the Australian Government Bureau of Meteorology, and the data are available from the Centre for Environmental Data Analysis (CEDA) Archive (Met Office, 2008). There are two main types of WPR in the Australian network: Stratospheric Tropospheric Profilers (STP) and Boundary Layer Profilers (BLP) (Dolman et al., 2018). Both operates at 55 MHz. The maximum detection heights of STP are 8 km for low mode and 20 km for high mode with a range resolution of 250 m and 500 m, respectively. For BLP, the maximum detection height for low mode is 7 km and 14 km for high mode, with a range resolution of 100 m and 250 m, respectively. Both types of WPR measurements have been validated and calibrated with radiosonde data, achieving the slope of the least-squares line of best fit close to 1 for both zonal winds and meridional winds and the random difference between WPR and radiosonde data of about 2 m s$^{-1}$ (Dolman et al., 2016). The wind vectors of WPR measurements from the CEDA archive are 30-minute averaged winds.

## 2.3 NWP model winds

In order to carry out the triple collocation analysis, predicted winds were extracted from the Aeolus Auxiliary Meteorological Data (AUX_MET) files. AUX_MET contains forecasted meteorological information at Aeolus observation locations (e.g. temperature and pressure) that is required for processing the L2B product (de Kloe et al., 2021). These meteorological parameters are generated by the fifth generation European Centre for Medium-Range Weather Forecasts (ECMWF) Integrated Forecast System (IFS) model. Predicted winds from the AUX_MET files, rather than model winds from analyses, are selected to avoid dependency between model analyses and observed winds during validation, as Aeolus winds used for validation have not yet been assimilated. In addition, although WPR measurements have been assimilated, the dependency of predicted winds

with WPR measurements becomes weaker with forecast time. As a result, these three data sets are assumed independent of each other, which is required for triple collocation analyses.

The ECMWF IFS model uses octahedral reduced Gaussian grid $T_{co}1279$ with the grid spacing of about 9 km at mid-latitudes. Vertically, there are 137 model levels. The effective spatial resolution in the free atmosphere of a model is usually 7-10 times the grid distance (Skamarock, 2004), so the effective resolution of the ECMWF IFS model is around 90 km in the free atmosphere. Although the effective spatial resolution may be higher in the planetary boundary layer (PBL) due to orographic forcing, the upper air where Aeolus mainly operates is generally uninformed. Therefore, in this study, we take about 90 km as the model effective resolution. It is noted that the AUX_MET extracts data from the ECMWF IFS model every 3 seconds along the Aeolus predicted ground track. With moving speed at around 7 km s$^{-1}$ with respect to the surface, each AUX_MET vertical profile is placed at an interval of about 21 km for a given off-nadir prediction (Rennie, 2021), which does not affect the effective resolution of the model. Regarding the data quality, the typical differences between radiosonde and ECMWF winds are 2-3 m s$^{-1}$ (Houchi et al., 2010).

## 2.4 Collocation criteria

To carry out error analyses, all data should be collocated both in time and in space. First, the nearest Aeolus L2B wind profiles were extracted based on their distance from WPR sites, which should be less than 75 km (Zhang et al., 2016; Guo et al., 2021). This is because many WPR sites in Australia are in coastal regions. The Aeolus ground tracks 100 km or more away from the WPR sites would be either over the ocean or inland. Thus, the wind difference caused by two different representative regions may have much impact on the inter-comparison analysis between Aeolus observations and WPR measurements, especially for the Mie-cloudy winds that are usually sampled at a lower level. Additionally, we would like to keep consistency with the existing Aeolus wind validation using radar profiler measurements, to enable comparison results easily. The validation for China from Guo et al. (2021) was the only study available when we carried out this work. Therefore, we chose the WPR sites within 75 km to the Aeolus profiles. The vector winds from WPR measurements with time closest to Aeolus observations were selected. The vector winds from AUX_MET were extracted from the profiles closest to each Aeolus L2B wind profile. Based on this criterion, there are 6 WPR sites available over Australia, shown in Fig. 1. Over the study period, there should be 5016 Aeolus data samples in total for each detection channel. After quality control based on the criteria in Section 2.1, there are 2171 and 394 data samples remaining, accounting for 43.28% and 7.85% of the Rayleigh and Mie wind measurements, respectively. The site information and available Aeolus data samples are summarised in Table 1.

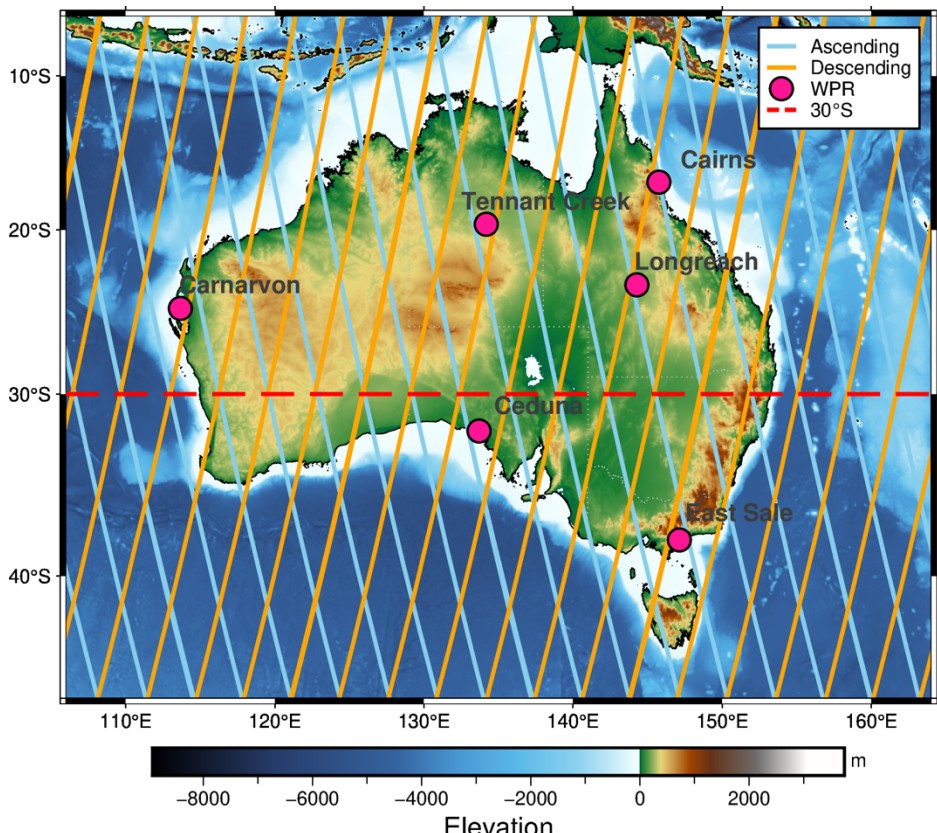

**Figure 1**. Location of wind profiling radars and Aeolus ground tracks over Australia. The pink marks on the map represent the sites of WPR used in this study, and the blue and orange lines indicate the Aeolus ground tracks for ascending and descending orbits, respectively. The red dashed line at 30°S is the boundary between tropics and extratropics. Shading with different colours represents the earth relief. The elevation data were accessed and the map was created by the authors using PyGMT (Wessel et al., 2019; General Bathymetric Chart of the Oceans (GEBCO) Compilation Group, 2021; Sandwell et al., 2021; Uieda et al., 2021).

**Table 1.** Information of ground-based WPR sites and Aeolus measurements

| Sites (Type) | Latitude (°) | Longitude (°) | Elevation (m) | Aeolus Ascending | | Aeolus Descending | |
|---|---|---|---|---|---|---|---|
| | | | | Overpass time (UTC) | Available samples (Rayleigh/Mie) | Overpass time (UTC) | Available samples (Rayleigh/Mie) |
| Longreach (STP) | -23.44 | 144.28 | 192 | 08:41 Thursday | 301/57 | 20:01 Wednesday | 320/15 |
| Carnarvon (STP) | -24.89 | 113.67 | 4 | - | - | 22:00 Friday | 368/13 |

| | | | | | | | | |
|---|---|---|---|---|---|---|---|---|
| Tennant Creek (STP) | -19.64 | 134.18 | 376 | - | | - | 20:40 Saturday | 151/35 |
| Cairns (BLP) | -16.95 | 145.75 | 4 | 08:31 Wednesday | 198/94 | | 20:00 Wednesday | 173/49 |
| East Sale (BLP) | -38.12 | 147.13 | 5 | 08:37 Thursday | 245/56 | | 19:40 Monday | 321/49 |
| Ceduna (BLP) | -32.13 | 133.70 | 15 | 09:32 Monday | 94/26 | | - | - |

165

Wind vectors from the WPR and AUX_MET data sets were converted to HLOS winds by using the following Eq. (1):

$$HLOS = -u_{Ref} \sin A - v_{Ref} \cos A \qquad (1)$$

where A is the azimuth angle of the Aeolus satellite, *Ref* represents either to WPR or AUX_MET. The geometry of Aeolus wind measurements is shown in Fig. 2.

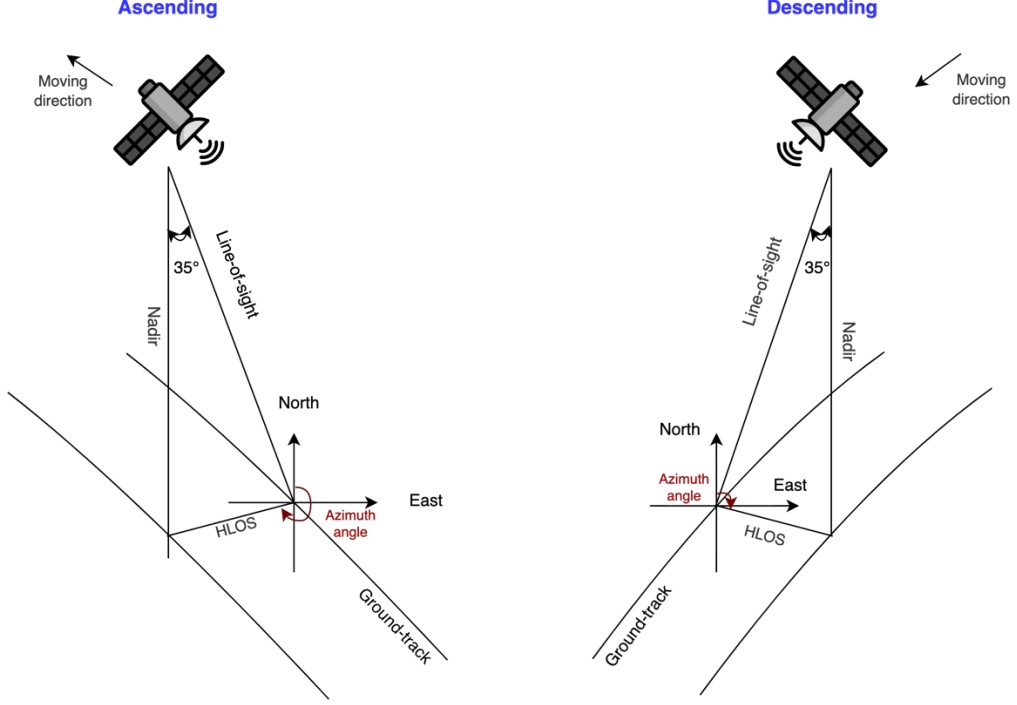

170

**Figure 2.** Geometry of Aeolus wind measurements.

Vertically, the HLOS winds from WPR and AUX_MET were converted to winds corresponding to Aeolus range bins by averaging the winds between the top and bottom heights of each vertical bin, shown in Fig. 3.

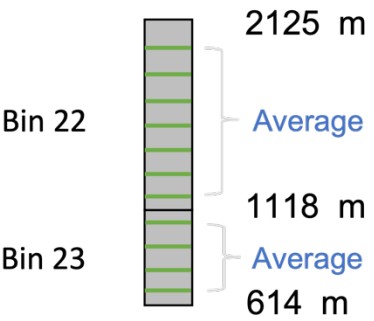

**Figure 3.** Sketch map of WPR and AUX_MET wind conversion to Aeolus range bins, where green bars represent the heights of WPR or AUX_MET winds.

## 2.5 Inter-comparison analysis

For inter-comparison analysis, we assumed that WPR is the ground truth. After data filtering and collocation, the mean bias (BIAS), standard deviation of wind difference (SD), scaled median absolute deviation (scaled MAD) and correlation coefficient (R) of both Rayleigh-clear winds and Mie-cloudy winds were quantified based on Eq. (2), (3), (4) and (5). Scaled MAD is used to represent random error because it is a robust statistic to measure data variability. When random errors are purely Gaussian distributed, scaled MAD is identical to SD; when distributions are not purely Gaussian, scaled MAD is less sensitive to outliers (Ruppert, 2011).

$$BIAS = \frac{1}{N} \sum_{i=1}^{N} \left( HLOS_{Aeolus,i} - HLOS_{Ref,i} \right) \tag{2}$$

$$SD = \sqrt{\frac{1}{N-1} \sum_{i=1}^{N} \left( \left( HLOS_{Aeolus,i} - HLOS_{Ref,i} \right) - BIAS \right)^2} \tag{3}$$

$$scaled\ MAD = 1.4826 \times median \left( \left| \left( HLOS_{Aeolus,i} - HLOS_{Ref,i} \right) - median \left( HLOS_{Aeolus,i} - HLOS_{Ref,i} \right) \right| \right) \tag{4}$$

$$R = \frac{\sum_{i=1}^{N} \left( HLOS_{Aeolus,i} - \overline{HLOS_{Aeolus}} \right) \left( HLOS_{Ref,i} - \overline{HLOS_{Ref}} \right)}{\sqrt{\sum_{i=1}^{N} (HLOS_{Aeolus,i} - \overline{HLOS_{Aeolus}})^2} \sqrt{\sum_{i=1}^{N} (HLOS_{Ref,i} - \overline{HLOS_{Ref}})^2}} \tag{5}$$

where subscript *Ref* represents WPR; N is the total number of data points; *i* is from 1, 2, 3…, N.

The confidence limits (uncertainty) for the biases are defined at a 95 % confidence interval. Since the distributions of wind differences are not always Gaussian, the confidence limits were estimated using the bootstrap method when the sample size is greater than 2.

Analyses were performed for all data, but also separated in ascending and descending orbits. In addition, errors as a function of height were also investigated. Since Aeolus observations over Australia have different vertical range bins settings for tropical and extratropical regions, we defined 12 new range bins based on the number of match-up samples in each range bin and the characteristics of the atmospheric circulation. Within the PBL and at a higher height, available match-up samples are limited. So, we defined several new groups at these heights by increasing the spacing. 500 hPa pressure surface is usually around 5.5 km above sea level, which is important for weather analysis and forecast, so we defined a group between 4.5 km and 6 km; the jet stream is usually from 8 km to 12 km in height, so two new groups were defined, which are 7.5 km - 10 km and 10 km -12.5 km (Wallace and Hobbs, 2006a). HLOS winds from their original range bins were grouped to defined range bins based on their centre of gravity (COG) heights. Moreover, to investigate the impact of range bin settings on error characteristics, we separated the results for tropics and extratropics.

## 2.6 Triple collocation analysis

To carry out triple collocation analysis, two other measurement systems are required besides Aeolus. In this study, they are WPR measurements and ECMWF IFS model equivalents. The temporal and spatial resolutions of these three systems are summarised in Table 2. We choose WPR measurement as the reference, which is system 1. Aeolus L2B winds and NWP winds are systems 2 and 3, respectively. All three systems are linearly correlated with the true HLOS winds, which can be described by Eq. (6), (7) and (8).

$$HLOS_1 = T + e_1 \tag{6}$$
$$HLOS_2 = a_2 + b_2T + e_2 \tag{7}$$
$$HLOS_3 = a_3 + b_3T + e_3 \tag{8}$$

where $T$ is the true value of HLOS winds; $a_i$ and $b_i$ are the intercept and the slope of the calibration for each system; $e_i$ is the random error of each system.

**Table 2.** Spatial and temporal resolution of the three data sets

|            | 1: WPR        | 2: Aeolus L2B                      | 3: AUX_MET      |
|------------|---------------|------------------------------------|-----------------|
| Horizontal | Point-based   | 87 km (Rayleigh) / 10-15 km (Mie)  | ~90 km          |
| Vertical   | 100/250/500 m | From 250 m to 2 km                 | 137 model levels |
| Temporal   | 30 minutes    | ~ 10 seconds / ~1-2 seconds        | Instantaneous   |

The method and equation derivation of triple collocation analysis are formulated in Vogelzang and Stoffelen (2012). To simplify the study, we assume the true measurement errors of each system are independent. Hence, no representation error contributes to the error covariances. Thus, the equations to calculate the error standard deviation of each system can be simplified to Eq. (9), (10) and (11).

$$\sigma_1 = \sqrt{\langle e_1^2 \rangle} = \sqrt{C_{11} - \frac{C_{12}\,C_{13}}{C_{23}}} \tag{9}$$

$$\sigma_2 = \sqrt{\langle e_2^2 \rangle} = \sqrt{C_{22} - \frac{C_{12}C_{23}}{C_{13}}} \tag{10}$$

$$\sigma_3 = \sqrt{\langle e_3^2 \rangle} = \sqrt{C_{33} - \frac{C_{23}C_{13}}{C_{12}}} \tag{11}$$

where $C_{ii}$ is the variance of each system, and $C_{ij}$ is the covariance between the system i and j; and $\langle\ \ \rangle$ represents the statistical averaging.

The calibration coefficients can be described by Eq.(12), (13), (14) and (15), and the calibration relations are shown in Eq.(16) and (17).

$$b_2 = \frac{C_{23}}{C_{13}} \tag{12}$$

$$b_3 = \frac{C_{23}}{C_{12}} \tag{13}$$

$$a_2 = \langle HLOS_2 \rangle - b_2 \langle HLOS_1 \rangle \tag{14}$$

$$a_3 = \langle HLOS_3 \rangle - b_3 \langle HLOS_1 \rangle \tag{15}$$

$$HLOS_2^* = \frac{HLOS_2}{b_2} - \frac{a_2}{b_2} \tag{16}$$

$$HLOS_3^* = \frac{HLOS_3}{b_3} - \frac{a_3}{b_3} \tag{17}$$

where $HLOS_2^*$ and $HLOS_3^*$ are the calibrated wind speed of system 2 and system 3.

**2.7 Wind variability analysis**

Wind observations are closely connected to the local atmospheric conditions. To investigate the influence of convection on Aeolus wind measurements, for every Rayleigh or Mie spatial sample, we employed the wind vectors from WPR measurements $\pm 2$ hours around the collocation points to quantify wind variability in each component and the turbulence kinetic energy (TKE), see Eq. (18), (19), (20) and (21). The results were averaged for Rayleigh and Mie channels, and the student t-test was performed.

$$Var(u) = \frac{\sum_{i=1}^{n}(u_i - \bar{u})^2}{n} \tag{18}$$

$$Var(v) = \frac{\sum_{i=1}^{n}(v_i - \bar{v})^2}{n} \tag{19}$$

$$Var(w) = \frac{\sum_{i=1}^{n}(w_i - \overline{w})^2}{n} \tag{20}$$

$$TKE = \frac{Var(u) + Var(v) + Var(w)}{2} \tag{21}$$

where $u_i$, $v_i$ and $w_i$ are WPR east-west, north-south and vertical winds at each time step (30 minutes) of ±2 hours (n=9) around the collocation points, and $\overline{u}$, $\overline{v}$ and $\overline{w}$ are the corresponding mean winds, respectively.

## 3 Results

After filtering out the invalid data and collocating all HLOS winds from the three data sets, results were derived from 1011 match-up samples of Rayleigh-clear winds and 224 match-up samples of Mie-cloudy winds.

### 3.1 Inter-comparison

The results of the inter-comparison analysis with WPR being the ground truth are summarised in Table 3 and presented in Fig. 4. From the scatter plots, it can be seen that the winds detected by the Rayleigh channel range from -40 m s⁻¹ to 60 m s⁻¹, while the wind speed from the Mie channel is lower, mainly ranging between -20 m s⁻¹ and 30 m s⁻¹. Overall, both Rayleigh-clear winds and Mie-cloudy winds are in good agreement with WPR measurements with R no less than 0.9 for all data. For Rayleigh-clear winds, the overall bias is -0.48 m s⁻¹ with a SD of 6.22 m s⁻¹ and a scaled MAD of 5.81 m s⁻¹. A larger bias (-0.71 m s⁻¹) was found during descending orbits, but no significant difference in random errors was detected during ascending and descending orbits. For Mie-cloudy winds, the bias for all data is 0.69 m s⁻¹, and the SD and the scaled MAD are 4.77 m s⁻¹ and 4.14 m s⁻¹, respectively. Moreover, the Mie channel has better performance on descending orbits (BIAS: -0.24 m s⁻¹; scaled MAD: 3.63 m s⁻¹) than ascending orbits (BIAS: 1.35 m s⁻¹; scaled MAD: 4.11 m s⁻¹).

**Table 3.** Results of inter-comparison with ground-based WPR measurements.

|  | Orbit | BIAS [m s⁻¹] | SD [m s⁻¹] | Scaled MAD [m s⁻¹] | R | N |
|---|---|---|---|---|---|---|
| Rayleigh-clear | All | -0.48 [-0.86, -0.09] | 6.22 | 5.81 | 0.92 | 1011 |
|  | Ascending | -0.06 [-0.73, 0.61] | 6.59 | 5.76 | 0.89 | 368 |
|  | Descending | -0.71 [-1.18, -0.26] | 5.99 | 5.73 | 0.88 | 643 |
| Mie-cloudy | All | 0.69 [0.08, 1.33] | 4.77 | 4.14 | 0.90 | 224 |
|  | Ascending | 1.35 [0.57, 2.19] | 4.76 | 4.11 | 0.86 | 132 |
|  | Descending | -0.24 [-1.23, 0.67] | 4.64 | 3.63 | 0.90 | 92 |

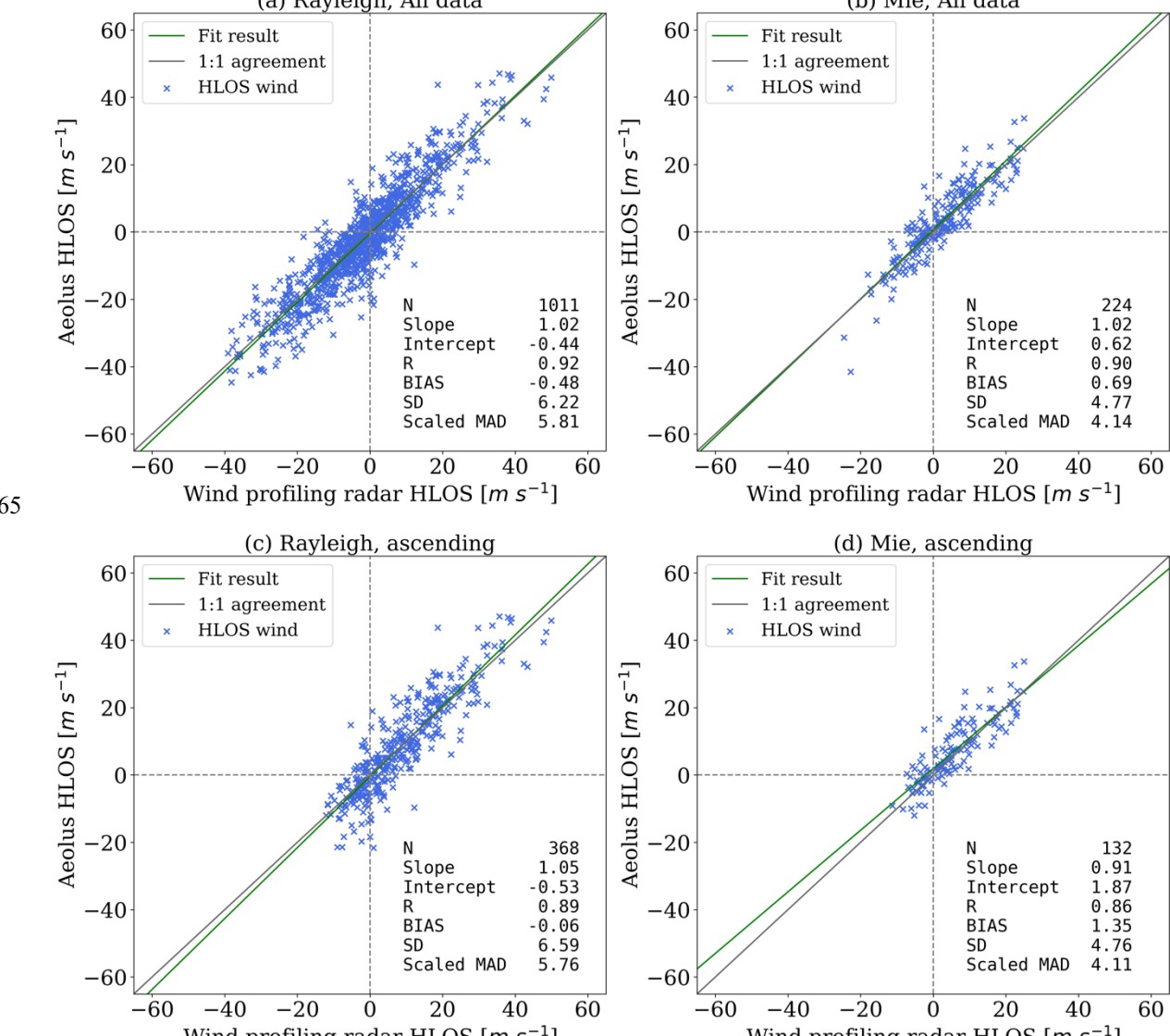

265

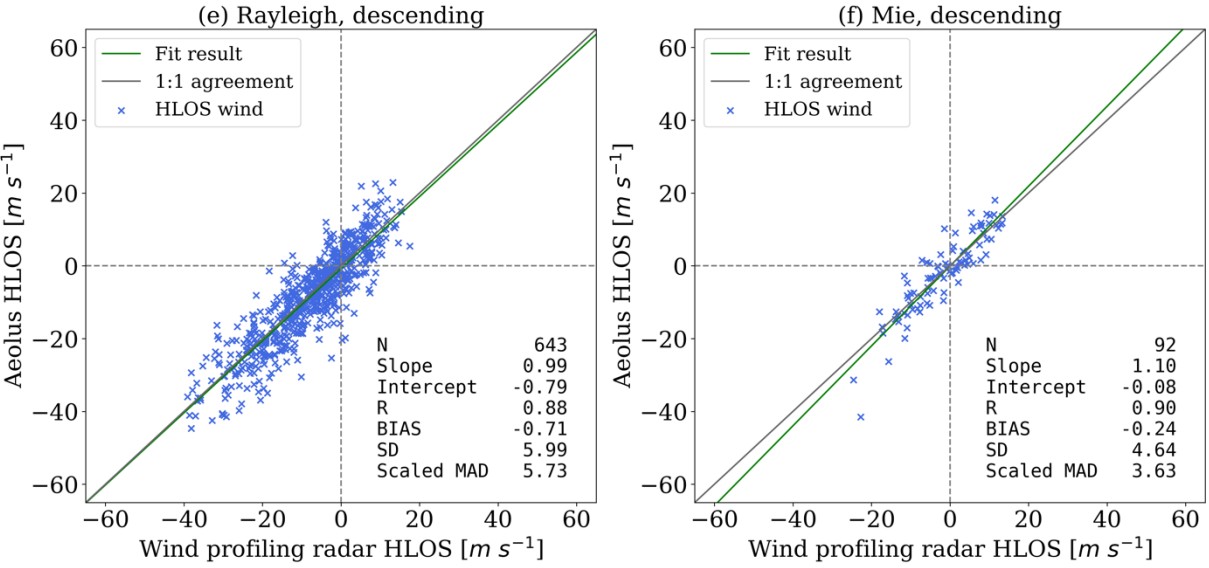

**Figure 4.** Scatter plots of Aeolus HLOS winds against WPR HLOS winds for all data, ascending orbits and descending orbits. Plot (a), (c) and (e) are for the Rayleigh-clear winds, and (b), (d) and (f) are for the Mie-cloudy winds. Green and grey lines indicate the fitted regression result and 1:1 agreement, respectively.

The wind difference as a function of height for all data (a), ascending (b) and descending (c) orbits is presented in Figure 5. Overall, there are more valid paired samples from the Rayleigh channel, except for heights below 750 m. Most of the Rayleigh-WPR samples distribute between 1,500 m and 20,000 m, while Mie-WPR samples mainly distribute below 10,000 m. Regarding the bias at each height, the Aeolus Rayleigh channel shows negative biases of about -1.7 m s$^{-1}$ between 750 m and 7,500 m for ascending orbits and about -0.8 m s$^{-1}$ between 1,500 m and 10,000 m for descending orbits, with the scaled MADs fluctuating at around 5 m s$^{-1}$. Above 10,000 m, for most heights, biases and scaled MADs become larger and/or more variable for Rayleigh wind match-ups. For the Mie channel, positive biases were detected between 750 m and 10,000 m with about 1.8 m s$^{-1}$ for ascending orbits and about 0.6 m s$^{-1}$ for descending orbits except for the height of 6,000 m – 7,500 m, and the scaled MADs are almost within 5 m s$^{-1}$. Negative biases and smaller scaled MADs were found below 750 m and above 10,000 m for both ascending and descending orbits for the Mie channel.

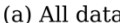
(a) All data

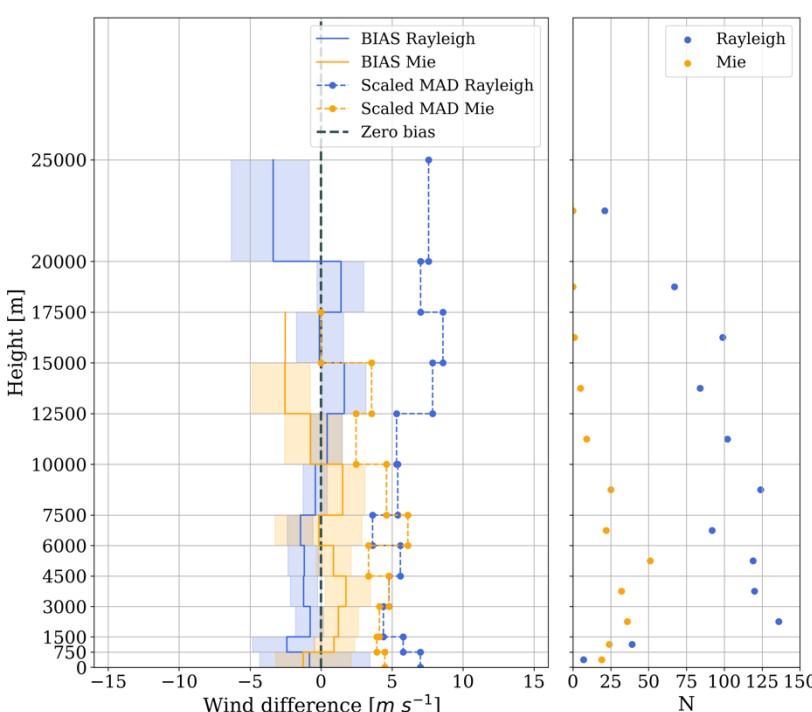

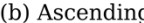

## (b) Ascending

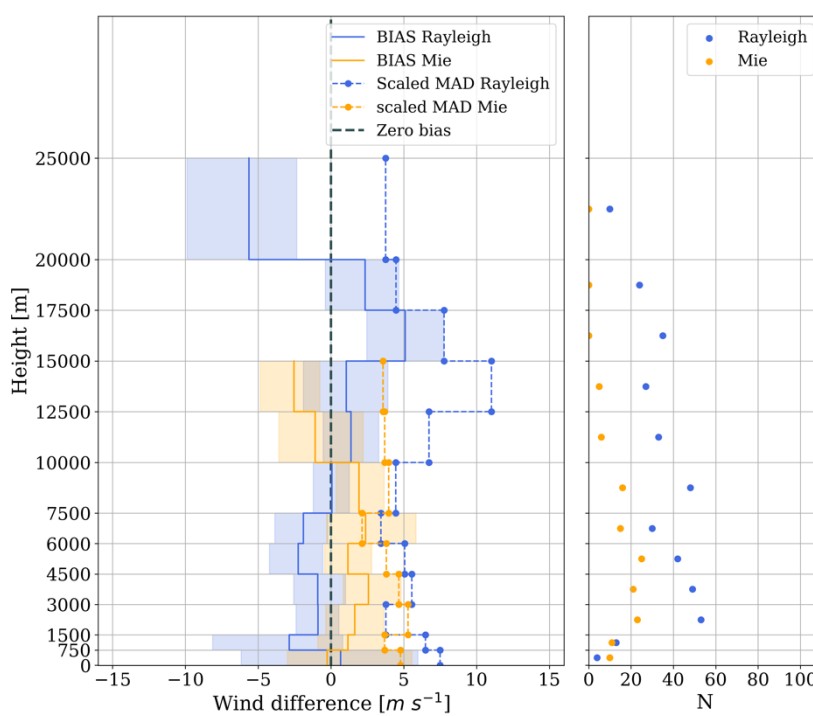

285

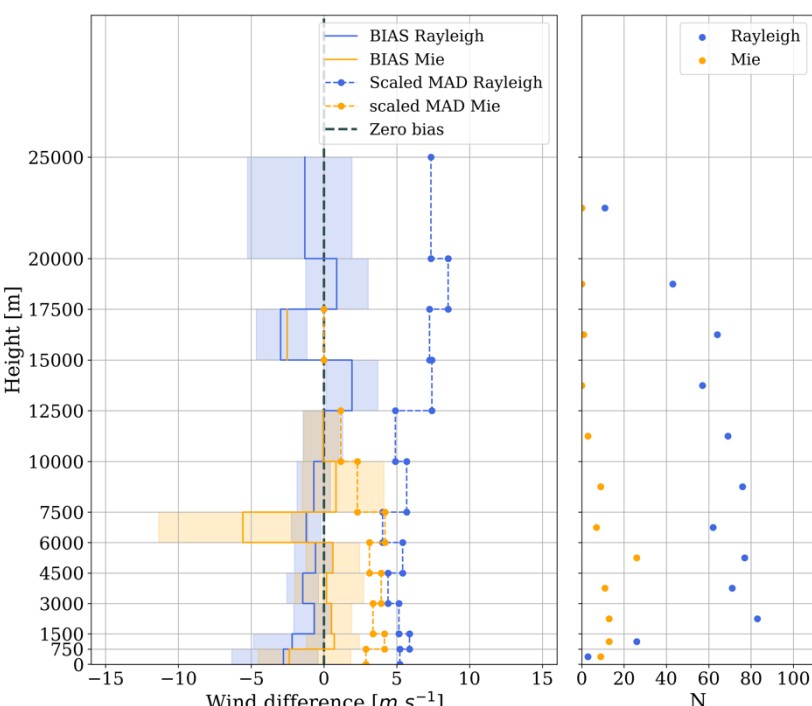

**(c) Descending**

**Figure 5.** Wind differences (Aeolus-WPR) with height for (a) all data, (b) ascending orbits and (c) descending orbits. Left: BIAS and scaled MAD of wind differences as a function of height with shading areas representing the uncertainty. Right: the number of available match-ups at each height. Blue and orange colours indicate the results for the Rayleigh and Mie channels, respectively.

To investigate the error characteristics for regions with different range bin settings, we separated the results from two regions, shown in Figure 6. For the tropics, larger biases from Rayleigh-clear winds and Mie-cloudy winds were found for the lower range bins with a thickness of 500 m. The random errors of Rayleigh-clear winds fluctuated at around 5 m s$^{-1}$ over the range bins of 1 km thickness, and the larger random errors were detected in range bins with a smaller thickness of 500 m or 750 m. For Mie-cloudy winds, the random errors for all range bins are lower than 5 m s$^{-1}$. For the sites over the extratropics, negative (positive) biases were found over most range bins for Rayleigh-clear (Mie-cloudy) winds. Random errors of Rayleigh-clear winds become smaller with height increasing, except for the range bin of 500 m thickness from higher heights, while the opposite is true for Mie-cloudy winds. The uncertainties of biases increase with height due to the limited number of match-up samples. Overall, based on Figure 6, smaller range bin thickness may contribute to larger random errors especially for Rayleigh-clear winds.

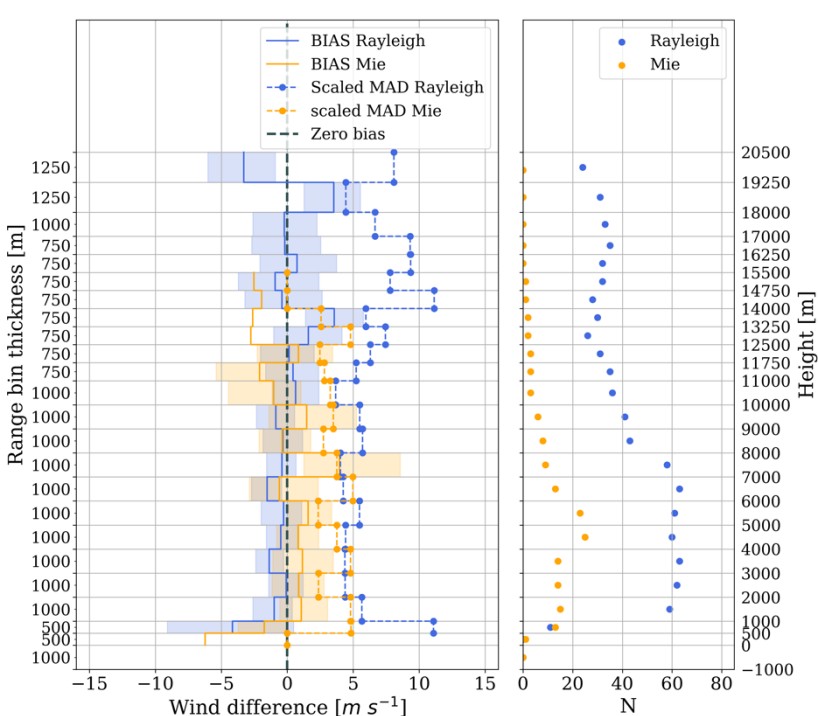

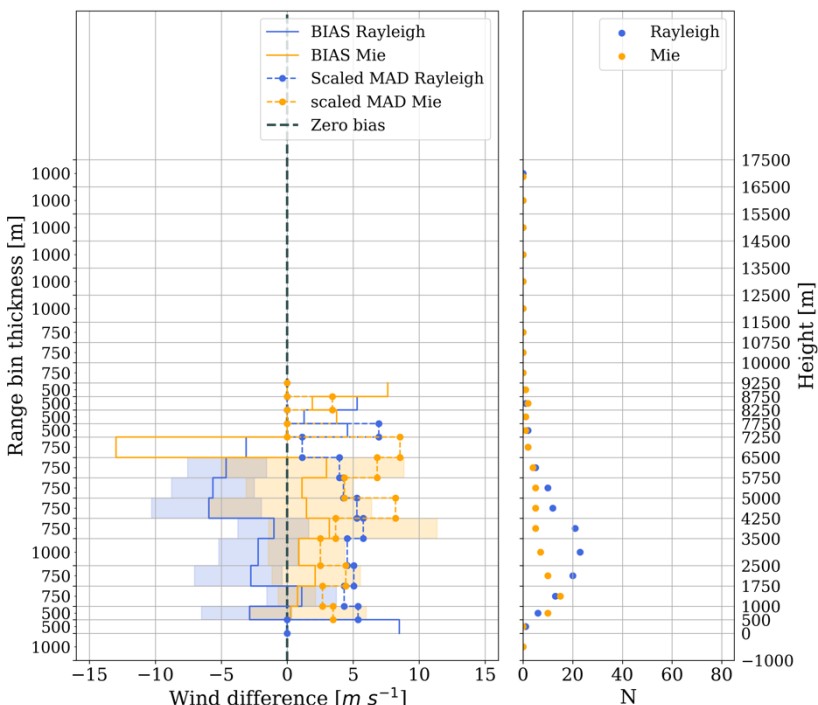

**Figure 6.** Wind differences (Aeolus-WPR) with range bins for (a) tropics and (b) extratropics. Left: distributions of BIAS and scaled MAD of wind differences over different range bins with shading areas representing the uncertainty. Right: the number of available match-ups at each range bin. Blue and orange colours indicate the results for the Rayleigh and Mie channels, respectively. Note: the heights on y-axis are just for reference, which are not exactly same with the actual heights of each vertical range bin.

## 3.2 Triple collocation

The result of the triple collocation analysis is shown in Table 4. For the combination of Rayleigh-clear winds, WPR and NWP model equivalents, the Aeolus measurements have the largest error standard deviation of 5.61 m s$^{-1}$ followed by WPR observations of 2.01 m s$^{-1}$. NWP model equivalent is most precise with an error standard deviation of 1.17 m s$^{-1}$. Similar results were also obtained from the combination with Mie-cloudy winds, and the error standard deviations are 3.50 m s$^{-1}$, 2.60 m s$^{-1}$ and 1.70 m s$^{-1}$ for Aeolus measurements, WPR observations and NWP model equivalents, respectively.

**Table 4.** Error standard deviation of three different systems

|  | 1: WPR [m s$^{-1}$] | 2: Aeolus L2B [m s$^{-1}$] | 3: AUX_MET [m s$^{-1}$] | N |
|---|---|---|---|---|
| **Rayleigh-clear** | 2.01 | 5.61 | 1.17 | 1011 |
| **Mie-cloudy** | 2.60 | 3.50 | 1.70 | 224 |

The calibration coefficients and relations for Aeolus L2B and NWP winds are shown in Table 5. For Rayleigh wind comparison, the Aeolus and the NWP model have similar patterns in wind estimation with intercepts of -0.404 m s$^{-1}$ and -0.236 m s$^{-1}$ and slopes of 1.044 and 1.033, respectively. Regarding the Mie wind comparison, the intercepts are 0.388 m s$^{-1}$ and 0.064 m s$^{-1}$ with the slopes of 1.106 and 1.075 for the Aeolus and the NWP model winds, respectively.

**Table 5.** Calibration coefficients of Aeolus L2B and NWP winds

|  | **2: Aeolus L2B** | | | **3: AUX_MET** | | |
|---|---|---|---|---|---|---|
|  | **a2** | **b2** | $\textbf{HLOS}_2^*$ | **a3** | **b3** | $\textbf{HLOS}_3^*$ |
| **Rayleigh-clear** | -0.404 | 1.044 | 0.958HLOS2 + 0.387 | -0.236 | 1.033 | 0.968HLOS3 + 0.228 |
| **Mie-cloudy** | 0.388 | 1.106 | 0.904HLOS2 - 0.351 | 0.064 | 1.075 | 0.930HLOS3 - 0.060 |

## 3.3 Wind variability

According to Table 6, all metrics of WPR wind variability for Mie-cloudy winds are higher than that of Rayleigh-clear winds, but only the difference in w wind component is statistically significant (p-value < 0.001). For Rayleigh wind detection, there is no big difference in wind variability during ascending and descending orbits, except for the w component. For Mie wind detection, wind variability (v, w and TKE) during ascending orbits is significantly higher than that during descending orbits, implying more convection in the late afternoon. Overall, the result suggests that the atmosphere may have larger variability during Mie-cloudy wind sampling, especially for ascending orbits.

**Table 6.** Results of wind variability based on WPR measurements

|  | **Var (u)** | **Var (v)** | **Var (w)** | **TKE** |
|---|---|---|---|---|
| **Rayleigh-clear [m$^2$ s$^{-2}$]** | 2.24 | 2.09 | 0.01 | 2.15 |
| **Mie-cloudy [m$^2$ s$^{-2}$]** | 2.48 | 2.45 | 0.03 | 2.48 |
| **p-value** | 0.358 | 0.181 | < 0.001 | 0.110 |
| **Rayleigh-clear Ascending [m$^2$ s$^{-2}$]** | 2.08 | 1.80 | 0.02 | 1.94 |
| **Rayleigh-clear Descending [m$^2$ s$^{-2}$]** | 2.34 | 2.25 | 0.01 | 2.26 |
| **p-value** | 0.298 | 0.070 | <0.001 | 0.097 |
| **Mie-cloudy Ascending [m$^2$ s$^{-2}$]** | 2.78 | 2.95 | 0.03 | 2.89 |
| **Mie-cloudy Descending [m$^2$ s$^{-2}$]** | 2.06 | 1.76 | 0.02 | 1.92 |
| **p-value** | 0.143 | 0.029 | 0.002 | 0.033 |

## 4 Discussion

The findings from the inter-comparison analysis indicate that both Rayleigh-clear and Mie-cloudy winds are in good agreement with the ground-based radar measurements with the biases for all data meeting the mission requirement of 0.7 m s$^{-1}$ (Ingmann

and Straume, 2016). However, the random errors represented by scaled MADs from both channels are larger than the specified random error of $< 3$ m s$^{-1}$ below 20 km (Ingmann and Straume, 2016), especially from the Rayleigh detection channel. These results are in line with many existing studies over different regions (Baars et al., 2020; Guo et al., 2021; Iwai et al., 2021; Chen et al., 2021). The large random errors are mainly because of unwanted signal losses in the instrument transmission and detection chain since the Aeolus launch (Krisch and Aeolus DISC, 2020), which impact the wind quality, especially for the Rayleigh channel. Regarding the performance during different orbit phases, a larger absolute mean bias was found for Rayleigh-clear winds during descending orbits, which is consistent with the results for the Northern Hemisphere (Martin et al., 2021), but the magnitudes of the biases ($< 1$ m s$^{-1}$) are smaller in this study. No big difference in random errors was found from Rayleigh-clear winds. For Mie-cloudy winds, a large mean bias (1.35 m s$^{-1}$) and random error (4.11 m s$^{-1}$) were identified during ascending orbits. One possible reason would be different representativeness conditions in the morning (descending) and afternoon (ascending). Figure 4 (d) and (f) show different wind distributions during ascending and descending orbits, hence suggesting a different circulation in the morning (descending) and afternoon (ascending). Moreover, Fig. 5 shows clouds peak at about 5 km height during descending orbits (at about 6:00 LT), while during ascending orbits, there are more uniform clouds from 2 km to 8 km height at the end of the day (at about 18:00 LT). In addition, Table 6 shows higher wind variability for ascending orbits during Mie-cloudy wind sampling, implying more convection in the late afternoon. So, the meteorological conditions during Mie wind measurements for the ascending and descending orbit phases appear quite different, which may imply different representativeness conditions and hence different random errors.

The vertical distributions of wind differences indicate that Mie-cloudy winds are more precise compared to Rayleigh-clear winds below 1,500 m for all data, which is consistent with the studies for China and Japan (Iwai et al., 2021; Wu et al., 2022). Higher random errors for Rayleigh-clear winds can partly be attributed to the smaller range bin thickness in the PBL. Below 750 m, large biases both for Rayleigh-clear and Mie-cloudy winds were found during descending orbits. This low accuracy may be related to the inhomogeneous topography at Cairns and at East Sale, which may have a larger impact on descending wind measurements, shown in Fig. 1. The large bias of Mie-cloudy winds between 6,500 m and 7,250 m in extratropical region mainly comes from the East Sale site during descending orbit on 30 November 2020. The WPR data quality on that day is not good with missing data on many range bins. To collocate with Aeolus wind at the COG height of 6,907 m (6,530 m-7,285 m), WPR winds should be converted by averaging the winds at the height of 6,755 m, 7,005 m and 7,255 m, but only wind at 6,755 m is available. After conversion, the HLOS$_{WPR}$ (-22.76 m s$^{-1}$) is much lower than the Aeolus measurements (-41.5 m s$^{-1}$). Moreover, the higher accuracies of Mie-cloudy winds during both ascending and descending orbits below 1,500 m suggest that the Mie channel is more capable of capturing winds within the PBL. This may also suggest the potential large impact of Mie-cloudy winds on data assimilation at these heights. In addition, during the descending orbit phase, for the Mie channel, the biases between 750 m and 6,000 m are smaller than 0.7 m s$^{-1}$, achieving the mission requirement.

The vertical distributions of available match-ups show most Rayleigh-WPR match-ups between 1,500 m and 20,000 m, since below 1,500 m less Rayleigh-clear winds are available due to the attenuation of the molecular signal because of Mie scattering within the PBL. Above 1,500 m, the number of available match-ups decreases with height. This is because the maximum measurement height of BLP is 7 km (10 km) and of STP is 8 km (20 km) for the low mode (high mode), so fewer data samples are available at higher heights. The majority of match-ups for Mie-cloudy winds distributes below 10,000 m, which is consistent with where Mie-scattering is expected to take place. Moreover, the number of Mie-WPR match-ups peaks between 4,500 m and 6,000 m during the descending orbits (about 6:00 LT) due to the mid-level clouds, such as altocumulus clouds that are mostly observed in warm spring and summer mornings (Gao et al., 2019).

The error standard deviations or random errors estimated by the triple collocation analysis for Rayleigh-clear winds (5.61 m s$^{-1}$) and Mie-cloudy winds (3.50 m s$^{-1}$) are roughly comparable with the results from inter-comparison analysis (5.81 m s$^{-1}$ for Rayleigh-clear winds and 4.14 m s$^{-1}$ for Mie-cloudy winds), indicating the Mie-cloudy winds are more precise than Rayleigh-clear winds. For the WPR, the temporal and spatial representation errors associated with the collocation criteria and the aggregated Aeolus observations are the main contribution to the random errors. The NWP random errors obtained are in line with expectations. Including the spatial representation error into account, Skamarock (2004) argues that the effective spatial resolution of a model in the free atmosphere is 7-10 times the grid distance, thus the horizontal resolution of Aeolus measurements for Rayleigh-clear winds and NWP model equivalents are almost alike, with the WPR resolving small-scale variance not detected by the Aeolus nor the NWP model. Given the coarse NWP resolution, the common variance of the coarse Rayleigh-clear winds and WPR will be small, and hence the impact of representativeness error on the Rayleigh comparison is limited. For the Mie comparison, the remaining common variance between the moderate-resolution Mie-cloudy winds and WPR is not resolved by the NWP model, which is coarsest, leading to the higher error standard deviation of NWP. Assuming a spatial representativeness error for NWP of 1 m s$^{-1}$ (Stoffelen et al., 2020), then the error standard deviations with respect to NWP become 1.37 m s$^{-1}$, 2.79 m s$^{-1}$ and 3.64 m s$^{-1}$ for NWP, WPR and Mie-cloudy winds, respectively. Compared with the results of triple collocation for Rayleigh winds, the random errors for NWP and WPR are higher. The study from Lin et al. (2016) explicitly showed that model wind accuracy near the ocean surface over convective areas is 4 times larger than that of clear areas. Moreover, based on the wind variability results in section 3.3, these higher values may be related to the vertical wind shear and convective conditions during Mie wind sampling. Additionally, the number of collocated samples for Mie comparison is just 224, which is much lower than the optimal amount (at least 1,000 samples) for triple collocation analysis, thus the results contain some uncertainty. When performing interpretation with respect to the system with the intermediate spatial resolution, that is the Mie scale, the spatial representation error represents part of the common resolved signal in WPR and Mie-cloudy winds, but the NWP model does not resolve this part of the signal. Thus, the error standard deviations with respect to the Mie scale become 1.97 m s$^{-1}$, 2.40 m s$^{-1}$ and 3.35 m s$^{-1}$ for NWP, WPR and Mie-cloudy winds, respectively, where the NWP winds still appear as the most precise.

This study is based on the Aeolus near real time 2B11 data. It is known that Mie-cloudy winds show systematic biases, for which a solution is in place for operational processing since July 2021 (Marseille et al., 2022). The needed correction for non-linearities of the Mie Spectral Response performs better when derived from an NWP based method than from in-orbit instrument calibration (Marseille et al., 2022). The L2B processing was adjusted accordingly. As a result, systematic biases
for moderate winds were removed and overestimation of strong winds was reduced. In order to evaluate whether these systematic errors impact the current validation results, we applied the correction method to the near real time 2B11 data during October 2020 – March 2021 to yield corrected Mie winds. The method of correction and validation results are shown in Appendix A. The analysis based on corrected Mie-cloudy winds suggest that the non-linearity bias correction has a potential to reduce the biases and random errors, especially for the samples from low to mid-level heights. These results do not affect
above discussion.

In addition, at the beginning of this study, we also tried the threshold values of 7 m s$^{-1}$ for Rayleigh-clear and 5 m s$^{-1}$ for Mie-cloudy winds for quality control referred to the study from Guo et al.(2021). The main results for inter-comparison analysis and triple collocation are summarized in Appendix B. We found that the threshold values obviously impact the number of
available data points. When we increase (decrease) the threshold value for Mie-cloudy (Rayleigh-clear) winds, more (fewer) data points become available. Regarding the statistics, the threshold values do not have much impact on the determined systematic and random errors for Rayleigh-clear winds that have around 1000 data points in total. For Mie-cloudy winds, the systematic and random errors are more sensitive to the threshold value partly because of fewer data points.

**5 Conclusions**

With the successful launch and operation of the Aeolus satellite, this study was undertaken to evaluate the Level-2B baseline 11 HLOS wind product during the Australian summer from October 2020 to March 2021. To achieve this, the Aeolus Rayleigh-clear and Mie-cloudy winds were inter-compared with ground-based WPR measurements. In addition, the triple collocation analysis was attempted for the combination of Aeolus winds (Rayleigh-clear and Mie-cloudy), WPR measurements and NWP model equivalents.

When comparing with the ground-based radar measurements, no obvious biases (absolute mean bias < 0.7 m s$^{-1}$) and good agreements ($R \geq 0.9$) were found for both Rayleigh-clear and Mie-cloudy winds for all match-up samples, but the bias for Mie-cloudy winds has a larger uncertainty. Moreover, the error characteristics are different between ascending and descending orbits. For the Rayleigh channel, the wind detection during ascending orbits has higher accuracy but larger uncertainty than
during descending orbits, while for the Mie channel, larger bias and random error were detected during ascending orbits. Vertically, the Mie channel was found to be more capable of detecting winds within the PBL, suggesting a larger impact of Mie winds in data assimilation at these heights. In addition, both difference statistics and triple collocation analysis showed

that Mie-cloudy winds are more precise than Rayleigh-clear winds. Moreover, triple collocation analysis showed that the NWP winds are most precise in representing Aeolus measurement scales, followed by WPR measurements, and Aeolus observations have the largest errors for both Rayleigh and Mie comparisons. Overall, the evidence from this study demonstrates that the space-borne lidar is able to detect winds with sufficient accuracy, which implies the potential benefit of Aeolus winds for data assimilation in numerical weather prediction, feeding different applications such as aeroplane route optimisation or wind energy prediction.

**Appendix A:**

The Mie-cloudy winds from 2B11 was corrected following Marseille et al. (2022). The corrected Mie-cloudy winds were sampled along the same profiles and range bins as the original 2B11 data and were extracted for further validation. The employed method is the same as for the original 2B11 Mie-cloudy winds.

After filtering out the outliers, there are 227 match-ups for analysis. Overall, the results from inter-comparison analysis are almost the same as the original Mie-cloudy winds with bias and scaled MAD becoming slightly smaller for all data (Table A1 and Fig. A1). However, for ascending orbits, the scaled MAD increased by 0.28 m s$^{-1}$. This may be caused by the low data quality over complex terrain at East Sale and Cairns. For descending orbits, both bias and scaled MAD increased somewhat, but not obvious.

**Table A1.** Results of inter-comparison with ground-based WPR measurements for corrected Mie-cloudy winds.

|  | Orbit | BIAS [m s$^{-1}$] | SD [m s$^{-1}$] | Scaled MAD [m s$^{-1}$] | R | N |
|---|---|---|---|---|---|---|
| Corrected Mie-cloudy | All | 0.67 [0.03, 1.31] | 4.90 | 4.10 | 0.89 | 227 |
|  | Ascending | 1.33 [0.52, 2.20] | 4.96 | 4.39 | 0.85 | 134 |
|  | Descending | -0.29 [-1.28, 0.61] | 4.68 | 3.72 | 0.90 | 93 |

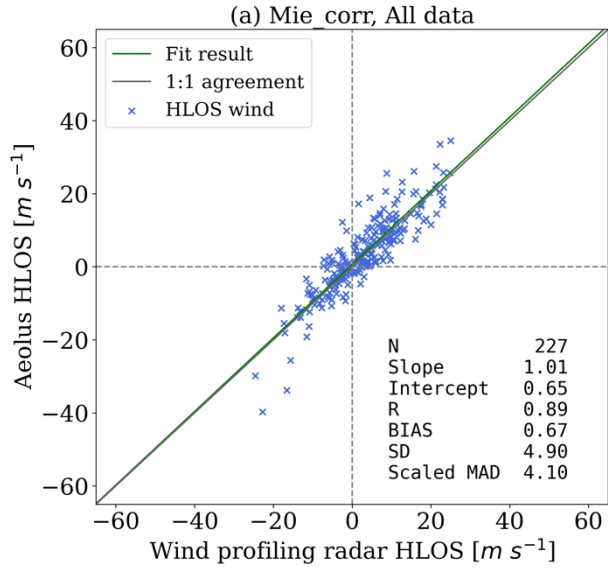

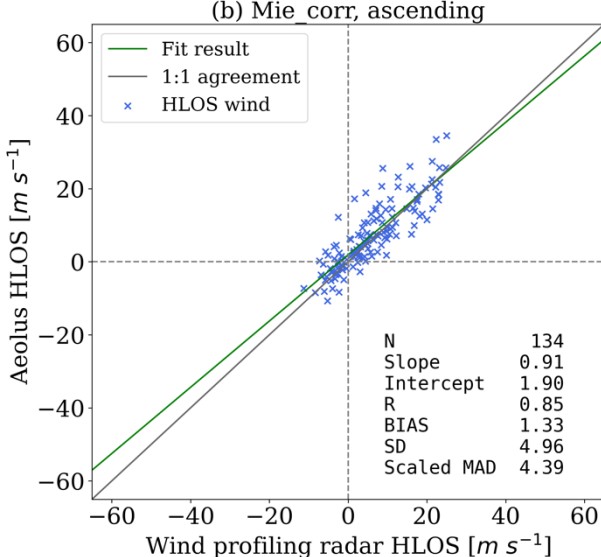

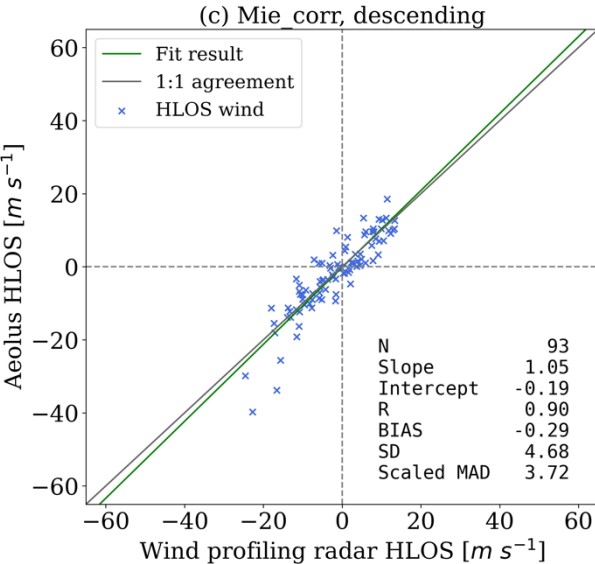

**Figure A1.** Scatter plots of corrected Mie-cloudy winds against WPR HLOS winds for all data (a), ascending orbits (b) and descending orbits (c). Green and grey lines indicate the fitted regression result and 1:1 agreement, respectively.

Regarding the wind difference as a function of height (Fig. A2), some improvements in accuracy and precision can be found mainly below 10,000 m. In particular, for data from all orbits, the random error below 3,000 m reduced by at least 0.6 m s$^{-1}$; the biases between 1,500 m and 6,000 m during descending orbits are close to 0. The reduction in random error can also be seen below 4,000 m and between 6,000 m and 8,000 m in tropics; the reduction in both biases and random errors can be found from 1,750 m to 3,500 m and from 4,250 m to 5,000 m in extratropics (Fig. A3). These are in line with the fact that most of the corrected Mie winds are from moderate wind speed range.

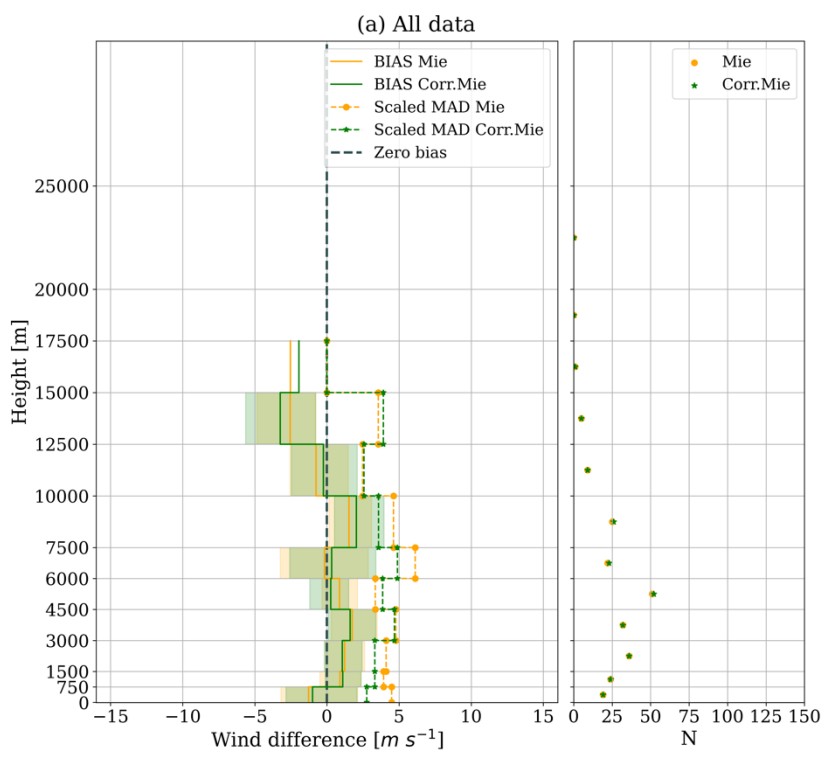

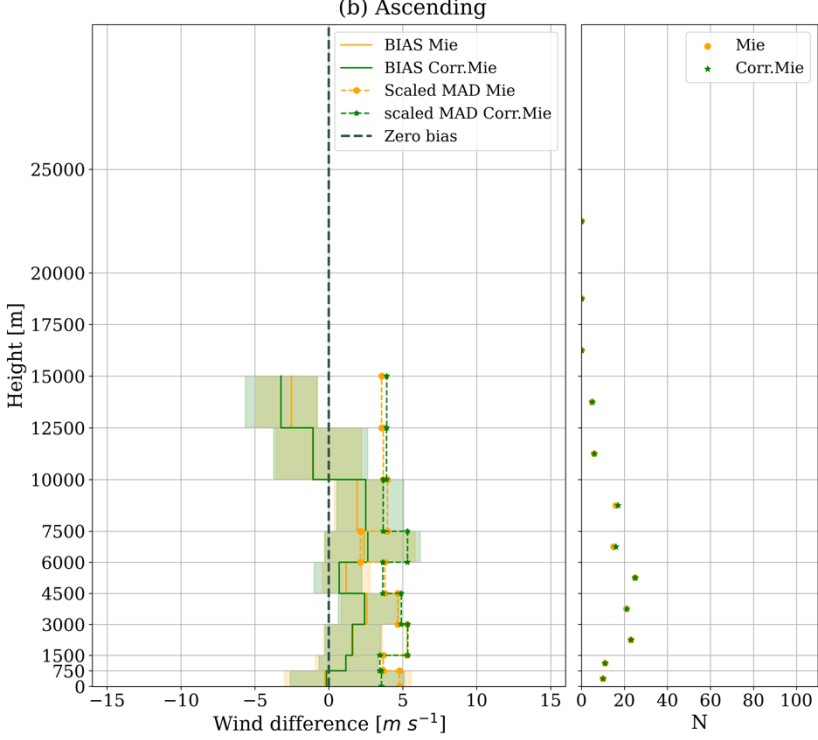

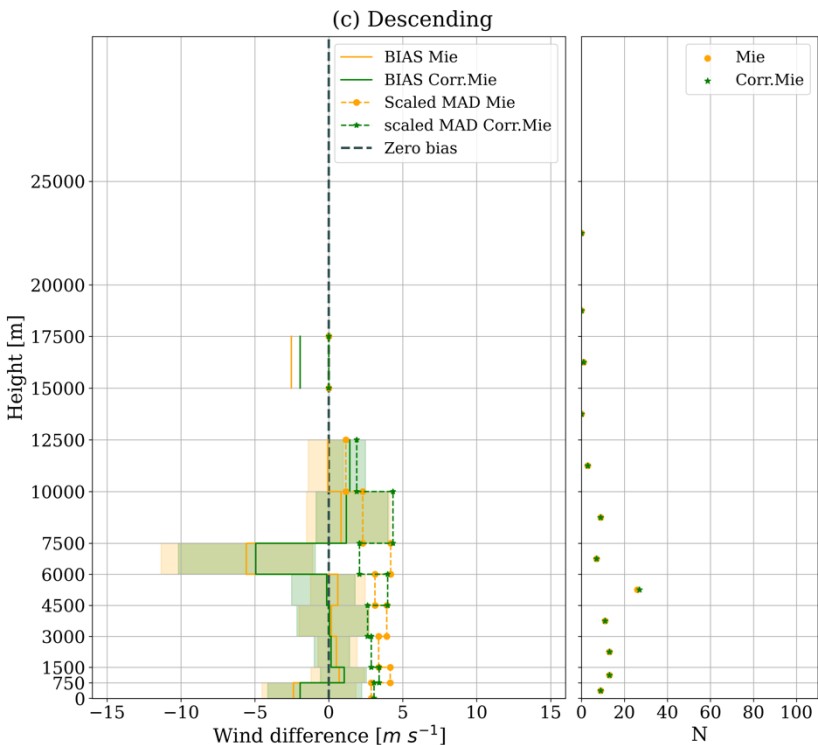

**Figure A2.** Wind differences (Aeolus-WPR) with height for (a) all data, (b) ascending orbits and (c) descending orbits. Left: BIAS and scaled MAD of wind differences as a function of height with shading areas representing the uncertainty. Right: the number of available match-ups at each height. Orange and green colours indicate the results for Mie-cloudy and corrected Mie-cloudy winds, respectively.

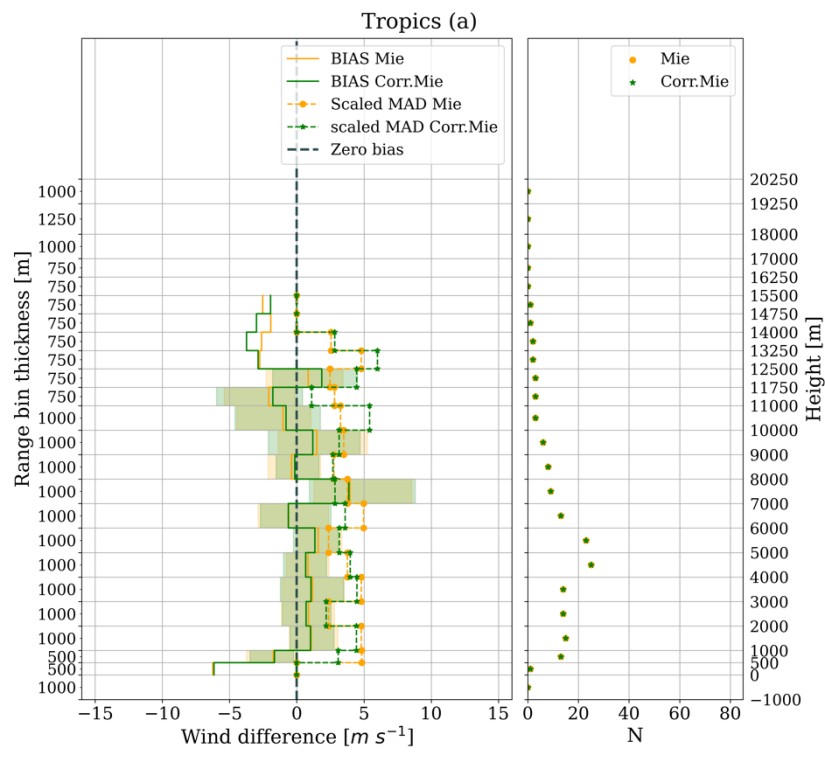

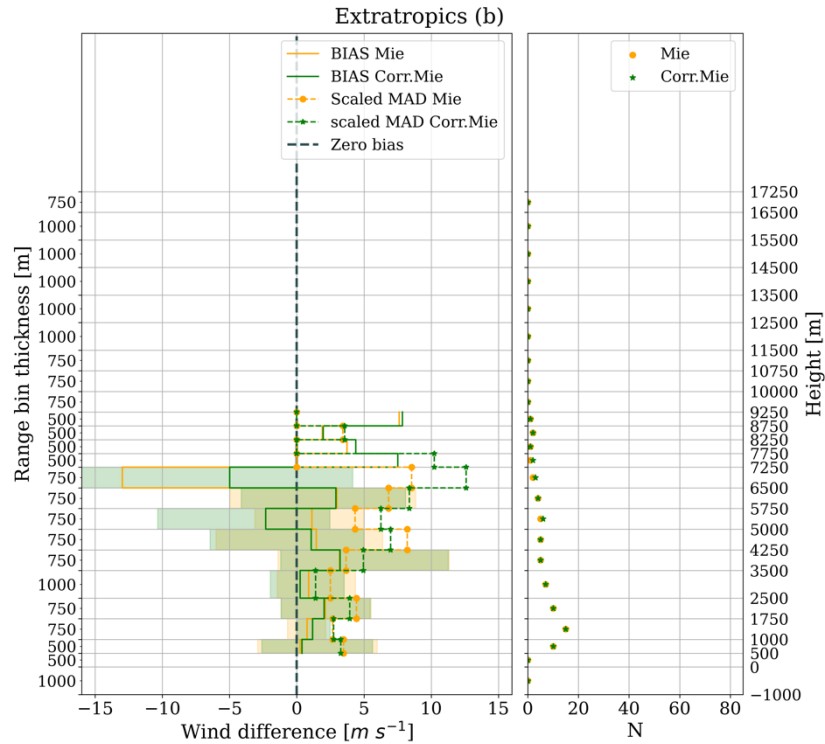

**Figure A3.** Wind differences (Aeolus-WPR) with range bins for (a) tropics and (b) extratropics. Left: distributions of BIAS and scaled MAD of wind differences over different range bins with shading areas representing the uncertainty. Right: the number of available match-ups at each range bin. Orange and green colours indicate the results for Mie-cloudy and corrected Mie-cloudy winds, respectively. Note: the heights on y-axis are just for reference, which are not exactly same with the actual heights of each vertical range bin.

The results from triple collocation analysis indicate that the correction can reduce the random error of Mie-cloudy winds to some extent but there is some uncertainty due to the limited number of collocated samples. The calibration coefficients are almost comparable with the results based on original 2B11 winds.

**Table A2.** Error standard deviation of three different systems

| | 1: WPR [m s$^{-1}$] | 2: Aeolus L2B [m s$^{-1}$] | 3: AUX_MET [m s$^{-1}$] | N |
|---|---|---|---|---|
| **Corrected Mie-cloudy** | 3.00 | 3.34 | 1.63 | 227 |

**Table A3.** Calibration coefficients of Aeolus L2B and NWP winds

| | 2: Aeolus L2B | | | 3: AUX_MET | | |
|---|---|---|---|---|---|---|
| | a2 | b2 | $HLOS_2^*$ | a3 | b3 | $HLOS_3^*$ |
| **Corrected Mie-cloudy** | 0.337 | 1.115 | 0.897HLOS2 – 0.302 | -0.031 | 1.129 | 0.886HLOS3 + 0.027 |

The results based on corrected Mie-cloudy winds suggest that the non-linearity bias correction has a potential to reduce the bias and random errors especially for the low to mid-level heights, which needs to be further demonstrated by enlarging the data samples or extending the study period.

**Appendix B:**

Regarding quality control, error estimate threshold values of 7 m s$^{-1}$ for Rayleigh-clear and 5 m s$^{-1}$ for Mie-cloudy winds were applied at the beginning of this study. The main results of inter-comparison and triple collocation analysis are summarised in Table B1 and B2.

**Table B1.** Results of inter-comparison with ground-based WPR measurements based on the error estimate threshold values of 7 m s$^{-1}$ for Rayleigh-clear and 5 m s$^{-1}$ for Mie-cloudy winds

| | Orbit | BIAS [m s$^{-1}$] | SD [m s$^{-1}$] | Scaled MAD [m s$^{-1}$] | R | N |
|---|---|---|---|---|---|---|
| Rayleigh-clear | All | -0.51 [-0.89, -0.13] | 6.10 | 5.80 | 0.92 | 998 |
| | Ascending | -0.17 [-0.81, 0.48] | 6.27 | 5.62 | 0.89 | 360 |

| | | | | | | |
|---|---|---|---|---|---|---|
| | Descending | -0.70 [-1.17, -0.23] | 6.00 | 5.76 | 0.88 | 638 |
| Mie-cloudy | All | 0.72 [0.06, 1.38] | 5.13 | 4.19 | 0.89 | 231 |
| | Ascending | 1.54 [0.69, 2.42] | 5.13 | 4.16 | 0.85 | 136 |
| | Descending | -0.46 [-1.49, 0.49] | 4.91 | 3.63 | 0.89 | 95 |

**Table B2.** Error standard deviation of three different systems based on the error estimate threshold values of 7 m s$^{-1}$ for Rayleigh-clear and 5 m s$^{-1}$ for Mie-cloudy winds

| | 1: WPR [m s$^{-1}$] | 2: Aeolus L2B [m s$^{-1}$] | 3: AUX_MET [m s$^{-1}$] | N |
|---|---|---|---|---|
| **Rayleigh-clear** | 2.01 | 5.51 | 1.17 | 998 |
| **Mie-cloudy** | 2.49 | 3.96 | 1.86 | 231 |

*Data availability.* Aeolus Level-2B11 wind product is available at ESA Aeolus Online Dissemination System (http://aeolus-ds.eo.esa.int/oads/access/, last access: 5 January 2022, ESA, 2021). The wind profiling radar measurements can be obtained from CEDA Archive (https://catalogue.ceda.ac.uk/uuid/9e22544a66ba7aa902ae431b1ed609d6, last access: 18 December 2021, Met Office, 2008). Aeolus AUX_MET files were created by the ECMWF centre and provided by the Royal Netherlands Meteorological Institute (KNMI). Earth relief data were accessed through PyGMT. (Wessel et al., 2019; GEBCO Compilation Group, 2021; Sandwell et al., 2021; Uieda et al., 505  2021).

*Author contributions.* All authors have made a contribution to the paper preparation. JK assisted in AUX_MET data preparation. GM conducted the Mie winds correction. HZ performed the data analysis and drafted the manuscript. AS, GM, CH and IK discussed the validation methods and helped interpreted the research findings. HZ revised the manuscript critically with the help from all co-authors.

*Competing interests.* Some authors are members of the editorial board of journal Atmospheric Measurement Techniques. The peer-review 510  process was guided by an independent editor, and the authors have also no other competing interests to declare.

*Acknowledgements.* This study is a part of the PhD project Aeolus Satellite Lidar for Wind Mapping that is a sub-project of the Innovation Training Network Marie Skłodowska-Curie Actions: Lidar Knowledge Europe (LIKE) supported by the European Union Horizon 2020 (Grant number: 858358). The authors thank ESA Aeolus Online Dissemination System (http://aeolus-ds.eo.esa.int/oads/access/) for access to Aeolus Level-2B baseline 11 near real time HLOS winds. We also acknowledge the Australian Government Bureau of Meteorology for 515  operating the WPR network over Australia and the CEDA for archiving and providing the WPR measurements. The authors are very thankful to KNMI for access to the AUX_MET data files and for being the secondment host institution. Our appreciation also goes to Michael Rennie for the information of NWP model equivalents.

*Financial support.* This research is a part of the PhD project Aeolus Satellite Lidar for Wind Mapping that is a sub-project of the Innovation Training Network Marie Skłodowska-Curie Actions: Lidar Knowledge Europe (LIKE) supported by the European Union Horizon 2020 520  (Grant number: 858358).

*Review statement.* This paper was edited by Oliver Reitebuch and reviewed by two anonymous referees.

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
