# Peer review of "Evaluation of Aeolus L2B wind product with wind profiling radar measurements and numerical weather prediction model equivalents over Australia"

_Atmospheric Measurement Techniques, 2022_

## Author Comment (AC1)

**Comment on amt-2022-63**

Anonymous Referee #1

Referee comment on "Evaluation of Aeolus L2B wind product with wind profiling radar measurements and numerical weather prediction model equivalents over Australia" by Haichen Zuo et al., Atmos. Meas. Tech. Discuss., https://doi.org/10.5194/amt-2022-63-RC1, 2022

We appreciate the constructive comments and suggestions by the reviewer, which are valuable for us to improve the study and the manuscript. All comments and concerns have been addressed item by item. The responses are highlighted in blue, and the changes are in orange below. Changes are also made within the manuscript accordingly.

**General Comments**

Overall well presented and useful comparison of Aeolus / wind profiles / NWP wind measurements

**Response:** We are pleased to hear these positive comments.

**Specific Comments :**

Bias' is discussed in several places ( e.g. Line 59 / Sect. 3.1, Table 3 ) but no confidence limits are given for these biases. This makes it impossible to understand if they are significant or if differences between 'bias' in different cases are significant. Please add confidence limits for the biases.

**Response:** Thank you very much for this suggestion. We have added the confidence limits for biases in Table 3. We also added a table (Table A1) in Appendix A to show the confidence limits for biases of corrected Mie-cloudy winds in the revised manuscript. The confidence limits are defined at a 95% confidence interval. Since the distributions of wind differences are not always Gaussian, confidence limits are estimated by using the bootstrap method.

**Table 1 (Table 3).** Results of intercomparison with ground-based WPR measurements.

|  | Orbit | BIAS [m s$^{-1}$] | SD [m s$^{-1}$] | Scaled MAD [m s$^{-1}$] | R | N |
|---|---|---|---|---|---|---|
| Rayleigh | All | -0.48 [-0.86, -0.09] | 6.22 | 5.81 | 0.92 | 1011 |
|  | Ascending | -0.06 [-0.73, 0.61] | 6.59 | 5.76 | 0.89 | 368 |
|  | Descending | -0.71 [-1.18, -0.26] | 5.99 | 5.73 | 0.88 | 643 |
| Mie | All | 0.69 [0.08, 1.33] | 4.77 | 4.14 | 0.90 | 224 |
|  | Ascending | 1.35 [0.57, 2.19] | 4.76 | 4.11 | 0.86 | 132 |
|  | Descending | -0.24 [-1.23, 0.67] | 4.64 | 3.63 | 0.90 | 92 |

**Table 2 (Table A1).** Results of intercomparison with ground-based WPR measurements for corrected Mie-cloudy winds.

|  | Orbit | BIAS [m s⁻¹] | SD [m s⁻¹] | Scaled MAD [m s⁻¹] | R | N |
|---|---|---|---|---|---|---|
| Corr. Mie | All | 0.67 [0.03, 1.31] | 4.90 | 4.10 | 0.89 | 227 |
|  | Ascending | 1.33 [0.52, 2.20] | 4.96 | 4.39 | 0.85 | 134 |
|  | Descending | -0.29 [-1.28, 0.61] | 4.68 | 3.72 | 0.90 | 93 |

In Figs 5 and A2, Fig. 5. 'uncertainty' in bias for different height bins is shown by shaded areas - these look surprisingly small given the very low number of samples in the height bins in many cases. How is 'uncertainty' defined ? 95% confidence limits or something else ?

**Response:** Thank you for these questions. The uncertainty was defined in Eq.(6) in the original manuscript.

$$Uncertainty = \frac{SD}{\sqrt{N}} \qquad (6)$$

where SD is the standard deviation of the biases with degrees of freedom of N-1, and N is the size of match-up samples. When there is only one match-up sample in a height bin, we did not calculate its uncertainty. This is why in some cases, the shading areas look extremely small.

In the revised manuscript, we re-defined the uncertainty (confidence limits) of biases at a 95% confidence interval, which was estimated by using the bootstrap method when the sample size is larger than 2. We re-generated all related plots, which can be seen in Figures 5 and 6 and Figures A2 and A3 in the revised manuscript.

(a) All data

(b) Ascending

[Figure]

(c) Descending

[Figure]

**Figure 1 (Figure 1).** Wind differences (Aeolus-WPR) with height for (a) all data, (b) ascending orbits and (c) descending orbits. Left: BIAS and scaled MAD of wind differences as a function of height with shading areas representing the uncertainty. Right: the number of available match-ups at each height. Blue and orange colours indicate the results for the Rayleigh and Mie channels, respectively.

**Tropics (a)**

[Figure]

**Extratropics (b)**

[Figure]

**Figure 2 (Figure 2).** Wind differences (Aeolus-WPR) with range bins for (a) tropics and (b) extratropics. Left: distributions of BIAS and scaled MAD of wind differences over different range bins with shading areas representing the uncertainty. Right: the number of available match-ups at each range bin. Blue and orange colours indicate the results for the Rayleigh and Mie channels, respectively.

[Figure]

[Figure]

**Figure 3 (Figure A1).** Wind differences (Aeolus-WPR) with height for (a) all data, (b) ascending orbits and (c) descending orbits. Left: BIAS and scaled MAD of wind differences as a function of height with shading areas representing the uncertainty. Right: the number of available match-ups at each height. Orange and green colours indicate the results for Mie-cloudy and corrected Mie-cloudy winds, respectively.

[Figure]

[Figure]

**Figure 4 (Figure A3).** Wind differences (Aeolus-WPR) with range bins for (a) tropics and (b) extratropics. Left: distributions of BIAS and scaled MAD of wind differences over different range bins with shading areas representing the uncertainty. Right: the number of available match-ups at each range bin. Orange and green colours indicate the results for Mie-cloudy and corrected Mie-cloudy winds, respectively.

The conclusions (Lines 361-364) say : "When comparing with the ground-based radar measurements, no significant biases (absolute mean bias < 0.7 m s⁻¹) and good agreements (R > 0.9) were found for both Rayleigh-clear and Mie-cloudy winds. For the Rayleigh channel, the wind detection during ascending orbits has higher accuracy than during descending orbits, while for the Mie channel, a large bias was obtained during ascending orbit. "

This says first there are 'no significant biases' and then 'a large bias was obtained'. Which is it ? Adding the confidence limits for the biases should help with getting this right.

**Response:** Thank you for your questions and suggestions. After adding the confidence limits, we have been able to improve the conclusions. The revised text can be seen in the second paragraph in Section 5 of the revised manuscript.

"When comparing with the ground-based radar measurements, no obvious biases (absolute mean bias < 0.7 m s⁻¹) and good agreements (R > 0.9) were found for both Rayleigh-clear and Mie-cloudy winds for all match-up samples, but the bias for Mie-cloudy winds has a larger uncertainty. Moreover, the error characteristics are different between ascending and descending orbits. For the Rayleigh channel, the wind detection during ascending orbits has higher accuracy but larger uncertainty than during descending orbits, while for the Mie channel, larger bias and random error were detected during ascending orbits."

**Minor points :**

There are numerous small grammar / language errors which are distracting - probably a copy editor can take care of most of these, although any co-authors who are proficient in English should also check.

We are very sorry for these small grammar mistakes. We tried our best to correct them, and the proofreading has been done carefully. We believe now the language is acceptable for publication.

For example : Lines 36-38

"Wind retrievals of ALADIN are based on light scattering by atmospheric molecules and particulates (aerosol, cloud droplets, and ice crystals) which move with the ambient wind and the Doppler effect (Ingmann and Straume, 2016). " - this says particulates ..move with ... the Doppler effect. It needs changing to

"Wind retrievals of ALADIN are based on light scattering by atmospheric molecules and particulates (aerosol, cloud droplets, and ice crystals), which move with the ambient wind, and on the Doppler effect (Ingmann and Straume, 2016). "

**Response:** We have corrected the sentence.

In Line 56 "Ray-clear " is used - everywhere else it is not shortened so it should be "Rayleigh-clear"

**Response:** We have fixed the error.

**Note:**

In addition to addressing all concerns from the anonymous reviewers, we re-plotted Figure A2 and added Figure A3 in Appendix A based on the Mie-cloudy winds (rather than Rayleigh-clear winds) and the corrected Mie- cloudy winds, thus making the improvement of Mie winds after correction more detectable.

---

## Author Comment (AC2)

**Response to comment on amt-2022-63**

Anonymous Referee #2

Referee comment on "Evaluation of Aeolus L2B wind product with wind profiling radar measurements and numerical weather prediction model equivalents over Australia" by Haichen Zuo et al., Atmos. Meas. Tech. Discuss., https://doi.org/10.5194/amt-2022-63-RC2, 2022

We are grateful to the reviewer for taking time to assess our manuscript. The constructive comments and suggestions are valuable for us to improve both our study and the manuscript. Below are the item-by-item responses (in blue) and the major changes (in orange) we made in the manuscript. The corresponding changes were also incorporated within the revised manuscript.

Since 22 August 2018, ESA's wind satellite Aeolus is circling the Earth at around 320 km altitude and capturing global wind profiles with its Doppler wind lidar ALADIN. To further improve the data quality and to use Aeolus observations in NWP models, the systematic and random errors must be understood. Besides global validations by means of NWP model comparison also regional/local validations with independent ground-based or in-situ reference measurements were already performed in several studies. This manuscript focuses on the validation of Aeolus wind measurements in the Australian domain with wind profiling radars as well as NWP model data and thus provides a useful contribution to the ongoing Aeouls Cal/Val activities.

The manuscript is well structured and written, presenting the obtained results with adequate figures. The paper deserves publication after some minor revisions.

**Response:** We appreciate this positive feedback.

**General comments**

For quality control, error estimate thresholds of 8 m/s for Rayleigh-clear and 4 m/s for Mie-cloudy winds are applied. Did you try other values and check how this affects the number of data points and the determined random and systematic error of Aeolus wind measurements?

**Response:** Yes, at the beginning of this study, we also tried the thresholds of 7 m/s for Rayleigh-clear and 5 m/s for Mie-cloudy winds for quality control referred to the study from Guo et al. (2021). The results for inter-comparison analysis and triple collocation are shown in brackets (orange) in Table 1 and 2.

The threshold values obviously impact the number of available data points. When we increase (decrease) the threshold value for Mie-cloudy (Rayleigh-clear) winds, more (fewer) data points become available. Regarding the statistics, the threshold values do not have much impact on the determined systematic and random errors for Rayleigh-clear winds that have around 1000 data points in total. For Mie-cloudy winds, the systematic and random errors are more sensitive to the thresholds partly because of fewer data points.

**Table 1 (Table B1).** Results of inter-comparison with ground-based WPR measurements.

| | Orbit | BIAS [m s$^{-1}$] | SD [m s$^{-1}$] | Scaled MAD [m s$^{-1}$] | R | N |
|---|---|---|---|---|---|---|
| Rayleigh | All | -0.48 [-0.86, -0.09] | 6.22 | 5.81 | 0.92 | 1011 |
| | | (-0.51 [-0.89, -0.13]) | (6.10) | (5.80) | (0.92) | (998) |
| | Ascending | -0.06 [-0.73, 0.61] | 6.59 | 5.76 | 0.89 | 368 |
| | | (-0.17 [-0.81, 0.48]) | (6.27) | (5.62) | (0.89) | (360) |
| | Descending | -0.71 [-1.18, -0.26] | 5.99 | 5.73 | 0.88 | 643 |
| | | (-0.70 [-1.17, -0.23]) | (6.00) | (5.76) | (0.88) | (638) |
| Mie | All | 0.69 [0.08, 1.33] | 4.77 | 4.14 | 0.90 | 224 |
| | | (0.72 [0.06, 1.38]) | (5.13) | (4.19) | (0.89) | (231) |
| | Ascending | 1.35 [0.57, 2.19] | 4.76 | 4.11 | 0.86 | 132 |
| | | (1.54 [0.69, 2.42]) | (5.13) | (4.16) | (0.85) | (136) |
| | Descending | -0.24 [-1.23, 0.67] | 4.64 | 3.63 | 0.90 | 92 |
| | | (-0.46 [-1.49, 0.49]) | (4.91) | (3.63) | (0.89) | (95) |

**Table 2 (Table B2).** Error standard deviation of three different systems

| | 1: WPR [m s$^{-1}$] | 2: Aeolus L2B [m s$^{-1}$] | 3: AUX_MET [m s$^{-1}$] | N |
|---|---|---|---|---|
| **Rayleigh-clear** | 2.01 (2.01) | 5.61 (5.51) | 1.17 (1.17) | 1011 (998) |
| **Mie-cloudy** | 2.60 (2.49) | 3.50 (3.96) | 1.70 (1.86) | 224 (231) |

We have incorporated these results in the last paragraph of Section 4 and added two tables showing the corresponding statistics in Appendix B within the revised manuscript.

A horizontal collocation radius of 75 km around the WPR sites was chosen. This is rather strict compared to other validation studies and recommendations which applied at least 100 km. The mentioned paper (Zhang et al.) focuses on aerosol comparison in the PBL were it can be quite variable. For wind this must not be the case. Did you try to increase the radius to a larger value (100 km or even higher) to check if this could improve the statistics by using more data points?

**Response:** Thank you for the question. We chose a horizontal collocation radius of 75 km mainly for two reasons. One reason is that many WPR sites in Australia are in coastal regions. The Aeolus ground tracks 100 km or more away from the WPR site would be either over the ocean or inland. Thus, the wind difference caused by two different representative regions may have much impact on the inter-comparison analysis between Aeolus observations and WPR measurements, especially for the Mie-cloudy winds that are usually sampled at a lower level. Another reason is that we would like to keep consistency with the existing Aeolus wind validation using radar profiler measurements, to enable comparison results easily. The validation for China from Guo et al. (2021) was the only study available when we carried out this work. Therefore, we did not increase the radius to a large value.

We have incorporated this point in Section 2.4 (L.145-151) within the revised manuscript.

Have the authors included the random errors of the WPR measurements as well as an estimate of the representativeness error in the determination of the Aeolus wind observation errors? Otherwise the determined random error of Aeolus would be a combination of different errors. Or can this be assessed from the triple collocation?

**Response:** Thank you for these questions.

For inter-comparison analysis, we did not take the random errors of the WPR measurements and the representativeness error of Aeolus winds into account. So, yes, the determined Aeolus random errors from inter-comparison analysis are the combination of different errors. This can also be demonstrated by the results from the triple collocation. If we take Rayleigh-clear winds as an example, it can be seen that the square root of the sum of variances for WPR and Aeolus $(\sqrt{2.01^2 + 5.61^2} = 5.96 \, m \, s^{-1})$ from triple collocation which agrees well with the scaled MAD ($5.81 \, m \, s^{-1}$) in Table 3 in the revised manuscript.

The impacts of spatial representativeness error on the results of triple collocation are discussed in paragraphy four in Section 4 (L. 381- 389 and L.395-399) in the revised manuscript.

"Including the spatial representation error into account, Skamarock (2004) argues that the effective spatial resolution of a model in the free atmosphere is 7-10 times the grid distance, thus the horizontal resolution of Aeolus measurements for Rayleigh-clear winds and NWP model equivalents are almost alike, with the WPR resolving small-scale variance not detected by the Aeolus nor the NWP model. Given the coarse NWP resolution, the common variance of the coarse Rayleigh-clear winds and WPR will be small, and hence the impact of representativeness error on the Rayleigh comparison is limited. For the Mie comparison, the remaining common variance between the moderate-resolution Mie-cloudy winds and WPR is not resolved by the NWP model, which is coarsest, leading to the higher error standard deviation of NWP. Assuming a spatial representativeness error for NWP of 1 m s[-1] (Stoffelen et al., 2020), then the error standard deviations with respect to NWP become 1.37 m s[-1], 2.79 m s[-1] and 3.64 m s[-1] for NWP, WPR and Mie-cloudy winds, respectively."

"When performing interpretation with respect to the system with the intermediate spatial resolution, that is the Mie scale, the spatial representation error represents part of the common resolved signal in WPR and Mie-cloudy winds, but the NWP model does not resolve this part of the signal. Thus, the error standard deviations with respect to the Mie scale become 1.97 m s[-1], 2.40 m s[-1] and 3.35 m s[-1] for NWP, WPR and Mie-cloudy winds, respectively, where the NWP winds still appear as the most precise."

Range bin thickness has an influence on the random error especially for Rayleigh wind measurements. Although the altitude dependence of the random and systematic errors was investigated, this was not mentioned or analyzed. The applied regrouping as well as the different range bin setting for tropics and extratropics covered by the domain hide this fact. It would be interesting to show this in the analysis, for example by separating the two different range bin settings areas.

**Response:** Thank you for this suggestion.

To investigate the impact of range bin thickness on error characteristics, we separated the results from tropics and extratropics, shown in Figure 1 and Figure 2. This new analyses have been incorporated into the revised manuscript, which can be found in Figure 6 and Figure A3. The added text can be found in Section 3.1 (L.293-302) and in Apendix A.

[Figure]

[Figure]

**Figure 1 (Figure 6).** Wind differences (Aeolus-WPR) with range bins for (a) tropics and (b) extratropics. Left: distribution of BIAS and scaled MAD of wind differences over different range bins with shading

areas representing the uncertainty. Right: the number of available match-ups at each range bin. Blue and orange colours indicate the results for the Rayleigh and Mie channels, respectively.

"To investigate the error characteristics for regions with different range bin settings, we separated the results from two regions, shown in Figure 6. For the tropics, larger biases from Rayleigh-clear winds and Mie-cloudy winds were found for the lower range bins with a thickness of 500 m. The random errors of Rayleigh-clear winds fluctuated at around 5 m s$^{-1}$ over the range bins of 1 km thickness, and the larger random errors were detected in range bins with a smaller thickness of 500 m or 750 m. For Mie-cloudy winds, the random errors for all range bins are lower than 5 m s$^{-1}$. For the sites over the extratropics, negative (positive) biases were found over most range bins for Rayleigh-clear (Mie-cloudy) winds. Random errors of Rayleigh-clear winds become smaller with height increasing, except for the range bin of 500 m thickness from higher heights, while the opposite is true for Mie-cloudy winds. The uncertainties of biases increase with height due to the limited number of match-up samples. Overall, based on Figure 6, smaller range bin thickness may contribute to larger random errors especially for Rayleigh-clear winds."

[Figure]

[Figure]

**Figure 2 (Figure A3).** Wind differences (Aeolus-WPR) with range bins for (a) tropics and (b) extratropics. Left: distributions of BIAS and scaled MAD of wind differences over different range bins with shading areas representing the uncertainty. Right: the number of available match-ups at each range bin. Orange and green colours indicate the results for Mie-cloudy and corrected Mie-cloudy winds, respectively.

"The reduction in random error can also be seen below 4,000 m and between 6,000 m and 8,000 m in tropics; the reduction in both biases and random errors can be found from 1,750 m to 3,500 m and from 4,250 m to 5,000 m in extratropics (Fig. A3)."

**Specific comments**

L.14: Please include the data set time period in the abstract in addition to the baseline.

**Response:** We added the time period in the sentence

'With the help of ground-based wind profiling radar measurements and the European Centre for Medium-Range Weather Forecasts (ECMWF) model equivalents, this study quantifies the error characteristics of Aeolus L2B (baseline-11) near real time horizontal line-of-sight winds across Australia during October 2020 – March 2021 by using both inter-comparison and triple collocation analysis.'

L.37: You could cite the ADM-Aeolus Science Report here (https://www.esa.int/About_Us/ESA_Publications/ESA_SP-1311_i_ADM-Aeolus_i)

**Response:** Thank you for the suggestion. We have changed the reference to the recommended report.

L.56: Change to Rayleigh-clear to be consistent

**Response:** We have fixed the error.

L.104: Please specify the latitude regions for both settings (30 deg S) to see which sites are affected by which range bin settings

**Response:** We added the latitude range for each region in Section 2.1 (L.102).

"Over Australia, there are two different range bin setting regions, i.e., the tropical setting region (30°S - 30°N) and the extratropical setting region (30°S - 60°S)."

In addition, we updated the map by adding a red dashed line at 30°S to show which sites are affected by which range bin settings.

[Figure]

**Figure 3 (Figure 1).** Location of wind profiling radars and Aeolus ground tracks over Australia. The pink marks on the map represent the sites of WPR used in this study, and the blue and orange lines indicate the Aeolus ground tracks for ascending and descending orbits, respectively. The red dashed line at 30°S is the boundary between tropics and extratropics. Shading with different colours represents the earth relief. The elevation data were accessed, and the map was created by the authors using PyGMT (Wessel et al., 2019; General Bathymetric Chart of the Oceans (GEBCO) Compilation Group, 2021; Sandwell et al., 2021; Uieda et al., 2021).

L.115: Change to WPR

**Response:** We have fixed the error.

L.160: How was the temporal collocation performed for WPR-comparisons? These have 30 min resolution. Did you average consecutive WPR-profiles?

**Response:** No, we chose the WPR measurements with time closest to Aeolus observations. We added this information in Secion 2.4 (L.151).

L.180: On what is the spacing of these new groups based?

**Response:** The spacing of these new groups is based on two main factors. One is the number of match-up samples in each range bin. Within the Planetary boundary layer (PBL) and at a higher height, available match-up samples are limited. So, we defined some new groups at these heights by increasing the spacing. The other factor is the characteristics of the atmospheric circulation. PBL is usually below 1.5 km, so we defined two new groups below 1.5 km; 500 hPa pressure surface is usually around 5.5 km above sea level, which is important for weather analysis and forecast, so we defined a group between 4.5 km and 6 km; the jet stream is usually around 8 km to 12 km, so we defined two new groups here (7.5 km - 10 km and 10 km -12.5 km) (Wallace and Hobbs, 2006).

We have added this explanation in Section 2.5 (L.197-203) in the revised manuscript.

L.195 Table 2: Where do the 90 km come from? You mentioned 3 seconds temporal resolution corresponding to about 21 km above.

**Response:** We are sorry for this confusion.

The model horizontal resolution for producing AUX_MET data is $T_{CO}1279$ with a grid spacing about 9 km at mid-latitudes. Based on the study from Skamarock (2004), the effective spatial resolution in the free atmosphere of a model is usually 7-10 times the grid distance, so the effective resolution of AUX_MET is around 90 km in the free atmosphere. The effective spatial resolution may be higher in the planetary boundary layer due to orographic forcing, but the upper air where Aeolus mainly operates is generally uninformed. Therefore, in this study, we take around 90 km as the model effective resolution.

The AUX_MET samples the ECMWF model every 3 seconds along the Aeulus ground track, making the spacing between each profile is 21 km, which does not affect the effective resolution of ECMWF model data.

We added this explanation in Section 2.3 (L.133-137) in the revised manuscript. We hope that it is now clearer.

L.238: Change to WPR

**Response:** We have fixed the error.

L.239: large -> larger

**Response:** We have fixed the error.

L.246 Fig.4: plot axes could be made symmetrical; change desending to descending (also in Appendix)

**Response:** We have re-plotted all figures, making the axes symmetrical and correcting the spelling mistakes.

[Figure]

[Figure]

**Figure 4 (Figure 4).** Scatter plots of Aeolus HLOS winds against WPR HLOS winds for all data, ascending orbits and descending orbits. Plot (a), (c) and (e) are for the Rayleigh-clear winds, and (b), (d) and (f) are for the Mie-cloudy winds. Green and grey lines indicate the fitted regression result and 1:1 agreement, respectively.

[Figure]

[Figure]

**Figure 5 (Figure A1).** Scatter plots of corrected Mie-cloudy winds against WPR HLOS winds for all data (a), ascending orbits (b) and descending orbits (c). Green and grey lines indicate the fitted regression result and 1:1 agreement, respectively.

L.262: As pointed out above, range bin thickness has an influence on the random error especially for Rayleigh-clear observations. This should be mentioned here.

**Response:** Thank you very much for this comment. That is a very good point. We have incorporated this point in Section 3.1 (L.295 and in L.301) within the revised manuscript.

"The random errors of Rayleigh-clear winds fluctuated at around 5 m s$^{-1}$ over the range bins of 1 km thickness, and larger random errors were detected in range bins with a smaller thickness of 500 m or 750 m."

"Overall, based on Figure 6, smaller range bin thickness may contribute to larger random errors, especially for Rayleigh-clear winds."

L.270 Fig.5c: Do you have an idea why there is a larger bias between 6 and 7.5 km?

**Response**: Thank you for pointing this out. We double-checked the raw data of WPR measurements and found the large bias between 6 km and 7.5 km mainly comes from the East Sale site during descending orbit on 30 November 2020. The WPR data quality on that day is not good with missing data on many range bins. To collocate with Aeolus winds at the height of 6907 m (6530 m-7285 m), WPR winds should be converted by averaging the winds at the height of 6755 m, 7005 m and 7255 m, but only wind at 6755 m is available. After conversion, the $HLOS_{WPR}$ (-22.76 m/s) is much lower than the Aeolus measurements (-41.5 m/s).

We have incorporated this explanation within the revised manuscript, which can be found in section 4 (L.357-361).

L.290 Table 6: second Var(u) --> Var(v)

**Response:** We have fixed the error.

L.290 Table 6: Does the wind variability has influence on the representativeness (random error) of Aeolus observations? For example, did you try to exclude times where the variability is high? Is the variability changing for ascending and descending orbits? (more convection for ascending orbits)

**Response:** Thank you for these questions. We added the variability information for ascending and descending orbits during Rayleigh and Mie sampling, respectively, in Table 6 in the revised manuscript. For Rayleigh winds detection, there is no significant difference in wind variability, except for the w component during ascending and descending orbits. For Mie wind detection, wind variability (v, w and TKE) during ascending orbits is significantly higher than that of during descending orbits. This implies more convection in the late afternoon, which may be related to the large systematic and random errors during ascending orbits for Mie-cloudy wind measurements. Due to the limited number of data points especially for Mie-cloudy winds, we did not try to remove the data with high wind variability.

**Table 3 (Table 6).** Results of wind variability based on WPR measurements ($m^2$ $s^{-2}$)

|  | Var (u) | Var (v) | Var (w) | TKE |
|---|---|---|---|---|
| **Rayleigh-clear** | 2.24 | 2.09 | 0.01 | 2.15 |
| **Mie-cloudy** | 2.48 | 2.45 | 0.03 | 2.48 |
| **p-value** | 0.358 | 0.181 | < 0.001 | 0.110 |
| **Rayleigh-clear Ascending** | 2.08 | 1.80 | 0.02 | 1.94 |
| **Rayleigh-clear Descending** | 2.33 | 2.25 | 0.01 | 2.26 |
| **p-value** | 0.298 | 0.070 | <0.001 | 0.097 |
| **Mie-cloudy Ascending** | 2.78 | 2.95 | 0.03 | 2.89 |
| **Mie-cloudy Descending** | 2.06 | 1.76 | 0.02 | 1.92 |

| p-value | 0.143 | 0.029 | 0.002 | 0.033 |
| --- | --- | --- | --- | --- |

We updated the text in Section 3.3.

"According to Table 6, all metrics of WPR wind variability for Mie-cloudy winds are higher than that of Rayleigh-clear winds, but only the difference in w wind component is statistically significant (p-value < 0.001). For Rayleigh wind detection, there is no big difference in wind variability during ascending and descending orbits, except for the w component. For Mie wind detection, wind variability (v, w and TKE) during ascending orbits is significantly higher than that during descending orbits, implying more convection in the late afternoon. Overall, the result suggests that the atmosphere may have larger variability during Mie-cloudy wind sampling, especially for ascending orbits"

We also added a sentence in L.347 in Section 4.

"In addition, Table 6 shows higher wind variability for ascending orbits during Mie-cloudy wind sampling, probably due to more convection in the late afternoon."

L.306: Why only at 5 km over such a long time period? What is the reason for this peak?

**Response:** We think that for Mie-cloudy winds, the number of data points probably peaks at 5 km during the descending orbits (about 6:00 LT) due to the mid-level clouds, such as altocumulus clouds that are mostly observed in warm spring and summer mornings (Gao et al., 2019).

We have added the possible reason in the third paragraph in Section 4 (L.372-374).

'Moreover, the number of Mie-WPR match-ups peaks between 4500 m and 6000 m during the descending orbits (about 6:00 LT) due to the mid-level clouds, such as altocumulus clouds that are mostly observed in warm spring and summer mornings (Gao et al., 2019).'

L.311: Smaller range bin thickness in the PBL region could also contribute to higher random errors

**Response:** Thank you for this comment. We have added this point into the Section 4, please see L.354.

'Higher random errors for Rayleigh-clear winds can partly be attributed to the smaller range bin thickness in the PBL.'

L.349: Please shortly summarize the improvements of these processor updates (non-linearities are already mentioned in the Appendix...). Are only Mie-cloudy observations affected or also Rayleigh-clear?

**Response:** We have added a short summary of the improvements of these processor updates in of Section 4 (L.401-410). Since the non-linear systematic biases are found in Mie winds, only Mie-cloudy winds were corrected, and no correction was applied to Rayleigh-clear winds.

'The needed correction for non-linearities of the Mie Spectral Response performs better when derived from an NWP based method than from in-orbit instrument calibration (Marseille et al., 2022). The L2B processing was adjusted accordingly. As a result, systematic biases for moderate winds were removed and overestimation of strong winds was reduced.'

'The analysis based on corrected Mie-cloudy winds suggest that the non-linearity bias correction has a potential to reduce the random errors, especially for the low to mid-level heights.'

**Note:**

In addition to addressing all concerns from the anonymous reviewers, we re-plotted Figure A2 and added Figure A3 in Appendix A based on the Mie-cloudy winds (rather than Rayleigh-clear winds) and the corrected Mie-cloudy winds, thus making the improvement of Mie winds after correction more detectable.

**Reference:**

Gao, C., Li, Y., and Chen, H.: Diurnal Variations of Different Cloud Types and the Relationship between the Diurnal Variations of Clouds and Precipitation in Central and East China, Atmosphere, 10, 304, https://doi.org/10.3390/atmos10060304, 2019.

GEBCO Compilation Group: GEBCO 2021 Grid, British Oceanographic Data Centre (BODC) [data set], 2021.

Guo, J., Liu, B., Gong, W., Shi, L., Zhang, Y., Ma, Y., Zhang, J., Chen, T., Bai, K., Stoffelen, A., de Leeuw, G., and Xu, X.: Technical note: First comparison of wind observations from ESA's satellite mission Aeolus and ground-based radar wind profiler network of China, Atmos. Chem. Phys., 21, 2945–2958, https://doi.org/10.5194/acp-21-2945-2021, 2021.

Marseille, G., Kloe, J., Marksteiner, U., Reitebuch, O., Rennie, M., and Haan, S.: NWP calibration applied to Aeolus Mie channel winds, Q.J.R. Meteorol. Soc., 1–15, https://doi.org/10.1002/qj.4244, 2022.

Sandwell, D. T., Goff, J. A., Gevorgian, J., Harper, H., Kim, S.-S., Yu, Y., Tozer, B., Wessel, P., and Smith, W. H. F.: Improved Bathymetric Prediction using Geological Information: SYNBATH, Earth Space Sci., 30, https://doi.org/10.1002/essoar.10508279.1, 2021.

Skamarock, W. C.: Evaluating Mesoscale NWP Models Using Kinetic Energy Spectra, Mon. Wea. Rev., 132, 3019–3032, https://doi.org/10.1175/MWR2830.1, 2004.

Uieda, L., Tian, D., Leong, W. J., Jones, M., Schlitzer, Toney, L., Grund, M., Yao, J., Magen, Y., Materna, K., Newton, T., Anant, A., Ziebarth, M., Wessel, P., and Quinn, J.: PyGMT: A Python interface for the Generic Mapping Tools, Zenodo, https://doi.org/10.5281/ZENODO.5607255, 2021.

Wallace, J. M. and Hobbs, P. V.: Atmospheric Science An Introductory Survey, 2nd ed., Elsevier, USA, 2006.

Wessel, P., Luis, J. F., Uieda, L., Scharroo, R., Wobbe, F., Smith, W. H. F., and Tian, D.: The Generic Mapping Tools Version 6, Geochem. Geophys. Geosyst., 20, 5556–5564, https://doi.org/10.1029/2019GC008515, 2019.